



**Exploring implications of input parameter uncertainties on GLOF modelling**
**results using the state-of-the-art modelling code, r.avaflow**
**Sonam Rinzin[1], Stuart Dunning[1], Rachel Joanne Carr[1], Ashim Sattar[2], Martin Mergili[3]**
**[1]School of Geography, Politics and Sociology, Newcastle University, United Kingdom**
**[2]Divecha Centre for Climate Change, Indian Institute of Science, India**
**[3]Institute of Geography and Regional Science, University of Graz, Austria**
Correspondence to: Sonam Rinzin at s.rinzin2@newcastle.ac.uk
**Abstract**
Modelling complex mass flow processes like glacial lake outburst floods (GLOFs) for hazard
and risk assessments involves substantial data and computational resources, often leading
researchers to use low-resolution, open-access data and parameters based on plausibility
rather than direct measurement, which, although effective in back analysis, introduces
significant uncertainties in forward modelling. To determine the sensitivity of the model outputs
stemming from input parameter uncertainties in the forward modelling, we selected nine
parameters relevant to GLOF modelling and performed a total of 78 simulations in the
physically-based r.avaflow model. Our results indicate that GLOF modelling outputs are
notably sensitive to six parameters, which are, in order of importance: 1) volume of mass
movements entering lakes; 2) DEM datasets; 3) the origin of mass movements; 4) mesh size;
5) basal frictional angle; and 6) entrainment coefficient. The volume of mass movement
impacting lakes has the greatest impact on GLOF output, with an average coefficient of
variation (CV) = 47%, while the internal friction angle had the least impact (CV=0.4%). We
recommend that future GLOF modelling should carefully consider the output uncertainty
stemming from the sensitive input parameters identified here, some of which cannot be
constrained before a GLOF and must be considered only statistically.



## 1 Introduction

Glacial lakes can store millions of cubic meters of water: as of 2015, it is estimated that glacial lakes (>=0.05 km$^2$) store about ~105.7 km$^3$ of water globally (Zhang et al., 2023a; Shugar et al., 2020; Zheng et al., 2021b). Although glacial lakes in High Mountain Asia (HMA) contribute only 4.6 km$^3$ to this total volume, they have experienced the greatest expansion (46%) between 1990 and 2018 (Shugar et al., 2020). Furthermore, over 28% of glacial lakes in the HMA are dammed by loose/destabilizing moraines (Fujita et al., 2013; Zheng et al., 2021b) and the majority of glacial lakes (70%) are exposed to mass inputs, in the form of ice/snow avalanches, rockfalls and landslides (Dubey et al., 2023). Although there is no substantial evidence for an increasing trend in glacial lake outburst floods (GLOF) within existing data (between 850 and 2022 CE) (Shrestha et al., 2023; Lützow et al., 2023; Veh et al., 2022; Veh et al., 2023), the GLOF frequency is expected to increase in the future (Zheng et al., 2021) because the glaciers and permafrost in HMA are extremely sensitive to rising temperatures (Gruber et al., 2017; Kääb et al., 2018). Meltwater resulting from the shrinkage of glaciers leads to the formation of new glacial lakes and the expansion of existing ones (Zhang et al., 2015; Wang et al., 2020). This process sometimes exposes them to mass movement from the slopes above and increases the total volume of stored water (Rounce et al., 2016). Additionally, the degradation of permafrost destabilizes the slopes surrounding the glacial lakes, increasing the likelihood of mass movements into lakes (Huggel, 2009).

Recent work has documented 3151 GLOF events between 850 and 2022 C.E. globally (Lützow et al., 2023) and 682 GLOF events in HMA between 1833 and 2022 (Shrestha et al., 2023). In the HMA alone, these GLOF events have resulted in 6907 human deaths, caused damage to more than 2200 buildings, 71 km$^2$ of agricultural land, 163 MW capacity of hydropower, 2000 livestock and numerous other structures, including bridges and roads (Shrestha et al., 2023). However, these reported deaths and damages are significantly underestimated because of patchy documentation (Carrivick and Tweed, 2016). Unfortunately, the risk from GLOF is expected to rise in the future with the anticipated expansion of glacial lakes (Zheng et al., 2021b; Zhang et al., 2023b) compounded by a growing population and the construction of structures in areas prone to GLOFs (Taylor et al., 2023; Nie et al., 2023).

Most GLOF events in HMA start with mass movements entering the lake from surrounding slopes, leading to the displacement of water and waves overtopping the dam (Shrestha et al., 2023; Lützow et al., 2023; Nie et al., 2018). Rock- or ice-avalanches and landslides entering the lake constitute 70% of known causes of HMA historical GLOF events (Shrestha et al., 2023). The overtopping waves cause moraine dam incision and dam failure, resulting in a sudden discharge of lake water. To a lesser extent, GLOF events are also triggered by factors



such as increased hydrostatic pressure from runoff snow and ice melt, intense rainfall and
cloud outbursts, and dam settling caused by the melting of ice cores or internal piping.  As the
flood propagates further downstream, it can transform into a debris flow and/ or a hyper-
concentrated flow/debris flood depending on the geologic and topographic characteristics of
the river channel (Gaphaz, 2017; Schneider et al., 2014; Westoby et al., 2015; Westoby et al.,
2014). These complex GLOF process chains are difficult to accurately capture in numerical
models, given the large number of processes and parameters, and the phase transformations
during the event, which limits our ability to model the impacts of the hazard cascade as a
whole.
**1.1 Numerical modelling of GLOFs**
Previous studies have used various modelling codes such as HEC-RAS (Sattar et al., 2021b),
BASEMENT (Worni et al., 2013; Worni et al., 2012; Byers et al., 2018), FLO-2D (Somos-
Valenzuela et al., 2015), RAMMS (Lala et al., 2018), and r.avaflow (Mergili et al., 2020b). Most
all these models, however, cannot model the evolution of the GLOF process chain through
interaction at the boundary of different processes involved (e.g. the interaction of mass
movements with the lake) and dynamic transformation of flow through entrainment and
deposition. To address this limitation, some of the studies modelled each component
separately and then fed the results of each modelling component into the next stage (Lala et
al., 2018; Schneider et al., 2014; Frey et al., 2018). For example, Lala et al (2018) have used
RAMMS to model mass movement from the surrounding slope into the lake, Heller–Hager and
BASEMENT to model wave propagation across the lake surface and BASEMENT to model
the subsequent downstream hydrodynamic evolution of GLOF. In contrast, the r.avaflow model
(Mergili et al., 2017; Mergili and Pudasaini, 2024) enables the integration of all components of
the GLOF process chain and their interactions and transformation without the need to combine
the results of different models. It enables the detailed modelling of the GLOF process chain,
covering everything from the initial trigger to the downstream propagation. r.avaflow is an
open-source, GIS-based tool for simulating mass flows over arbitrary terrain. Furthermore,
r.avaflow is open source and allows modification of all input parameters, making it suitable for
conducting GLOF parameter sensitivity analysis (Mergili et al., 2017; Mergili and Pudasaini,

89  2024).

r.avaflow utilizes the total variation diminishing non-oscillatory central differencing (NOC-TVD)
numerical scheme (Wang et al., 2004) to solve an enhanced version of the Pudasaini multi-
phase flow model (Pudasaini and Mergili, 2019). It also offers added features for entrainment,
deposition, dispersion, and phase transformation. Because of these features, r.avaflow can
model the full process chain of a GLOF and flow transformation due to erosion of bed material



and deposition of entrained material (Mergili et al., 2017; Mergili and Pudasaini, 2024).
However, the precision of this model output depends on the accuracy of various input
parameters and initial conditions, including the release height of mass, the resolution and
vertical accuracy of the digital elevation model (DEM), density, entrainment, and frictional
parameters (Mergili et al., 2017). The difficulty involved in getting accurate measurements of
these parameters introduces substantial uncertainty in the modelling results.
Because of the significant logistic challenges associated with collecting field data and the
financial costs involved in acquiring high-resolution remote sensing data, many of the
parameters in GLOF modelling are derived from open-access data, leading to considerable
uncertainties in the resultant discharge, inundation extent, and arrival times. Also, certain
factors such as the volume of mass movement entering the lake are impossible to measure
accurately before a GLOF event. For example, the global-scale DEM, SRTM GL1, with a
ground resolution of 30 m, is commonly employed in GLOF modelling without adequately
considering the inherent uncertainty due to horizontal and vertical inaccuracies in this DEM
(Rinzin et al., 2023). Similarly, the origin of avalanches and other mass movements is
determined using low to medium-resolution remote sensing imagery and DEM, often
supplemented by secondary datasets like permafrost data (Obu et al., 2019), which can
introduce notable uncertainties (Sattar et al., 2023; Allen et al., 2016). When estimating the
volume of avalanches entering lakes, DEM differencing between pre- and post-event
conditions can be advantageous for reconstructing historical events (Baggio et al., 2021;
Zheng et al., 2021a), although the accuracy is contingent upon the vertical and horizontal
accuracy and resolution of the data, and the temporal interval between data accusation.
Likewise, when ice is considered the sole source of avalanches, ice thickness is employed to
calculate the avalanche volume (Allen et al., 2022), for which the accuracy of computed
volume relies on the resolution and availability of data in the region of interest. Under the
circumstances when the depth of landslides and avalanches are not known, conservative
thicknesses of 10, 30, and 50 m based on past events (Dubey et al., 2023) are often utilised
for forward modelling, further contributing to significant uncertainties in the modelling results
(Rounce et al., 2017; Rounce et al., 2016; Dubey and Goyal, 2020).
Moreover, the flow parameters in r.avaflow are adjusted and optimised based on the fit of the
model's results to well-documented past events (Mergili et al., 2017; Mergili et al., 2020a; Vilca
et al., 2021) and the physically plausible range suggested by Mergili et al. (2017), Mergili et
al. (2018b) and Mergili et al. (2018a). Efforts to fine-tune parameters to fit with historical events
of varying magnitude, temporality and spatiality have led to the use of wide-ranging values.
For example, Mergili et al. (2020b) used an internal solid friction angle of 28° to reconstruct



the 1941 GLOF process chain of Lake Palcacocha in the Cordillera Blanka, Peru. In contrast,
Vilca et al. (2021) used 45° to model the 2020 glacial lake outburst process chain of Lake
Salkantycocha located in Cordillera Vilcabamba of Peru. Likewise, the value of the basal
friction angle ranges between 6-18° (Baggio et al., 2021; Mergili et al., 2020a) (Supplementary
Figure 1 (Fig. S1)). Because each GLOF event is inherently distinct, even when originating
from the same glacial lake (Emmer and Cochachine, 2013; Lala et al., 2018), employing
reconstructed values from past events for forward modelling introduces substantial
uncertainties (Gaphaz, 2017; Mergili et al., 2020b). Finally, r.avaflow model outputs are
extremely sensitive to parameters like entrainment coefficient value, basal friction angle and
initial release volume (Mergili et al., 2018b; Mergili et al., 2018a; Baggio et al., 2021). However,
to our knowledge, how changes in the values of these input parameters affect the model output
(for example, peak and total flow, flow depth, flow velocity and arrival time) is not known.
To determine the relative contribution of uncertainties in different input parameters to variability
in GLOF extent, we identified nine out of 38 input parameters and initial conditions relevant to
GLOF flow modelling that have been previously identified as the most important in the
literature: digital elevation model; mesh size; the volume of mass movement impacting the
lake; the origin of mass movement impacting the lake; grain density of mass movement
impacting the lake; entrainment coefficient; internal friction angle; basal friction angle; and,
fluid friction number (Table S1). We assessed the sensitivity of the model output to each of
these parameters by conducting up to 10 r.avaflow simulations per parameter and varying
their values within the range determined from the literature that employed r.avaflow modelling
(Fig. S1). We investigated the impact of variation in these parameter values on the model
outputs and used the following diagnostic variables: peak discharge; total discharge; flow
arrival time; flow height; flow velocity and reach distance. We then calculated the coefficient
of variation for each parameter and ranked them based on this metric.
**2 Study site**
Here, our sensitivity analysis is conducted on Thorthormi Tsho located at 28.10° N, 90.27° E
in the Lunana region of the Bhutan Himalaya (Fig. 1). The area of Thorthormi Tsho has
expanded by ~192% since 1990, evolving into the largest proglacial lake (area = 4.35 km$^2$)
in Bhutan by 2020 (Rinzin et al., 2023) (Fig. 1B and 1E). Although the lake level was lowered
by 5m by artificially draining out the water between 2008 and 2012 (Nchm, 2019a),
Thorthormi Tsho is marked as the most dangerous glacial lake (Nchm, 2019a; Rinzin et al.,
2021) (Fig. 1B). In recent years, Thorthormi Tsho has produced two GLOF events (Nchm,
2023); the first one occurred on June 20, 2019 (Nchm, 2019b), the latest on October 30,
2023. Also, modelling of future predicted GLOF from Thorthormi Tsho shows it can produce





a flood with flow volume up to 300 × 10⁶ m³ of water with a peak discharge of up to 75000
m³ s⁻¹, affecting more than 16000 people and various infrastructures downstream of this
glacial lake (Rinzin et al., 2023). This high outburst susceptibility and potential make
Thorthormi Tsho an ideal candidate for GLOF modelling to improve our modelling output
GLOF uncertainty.

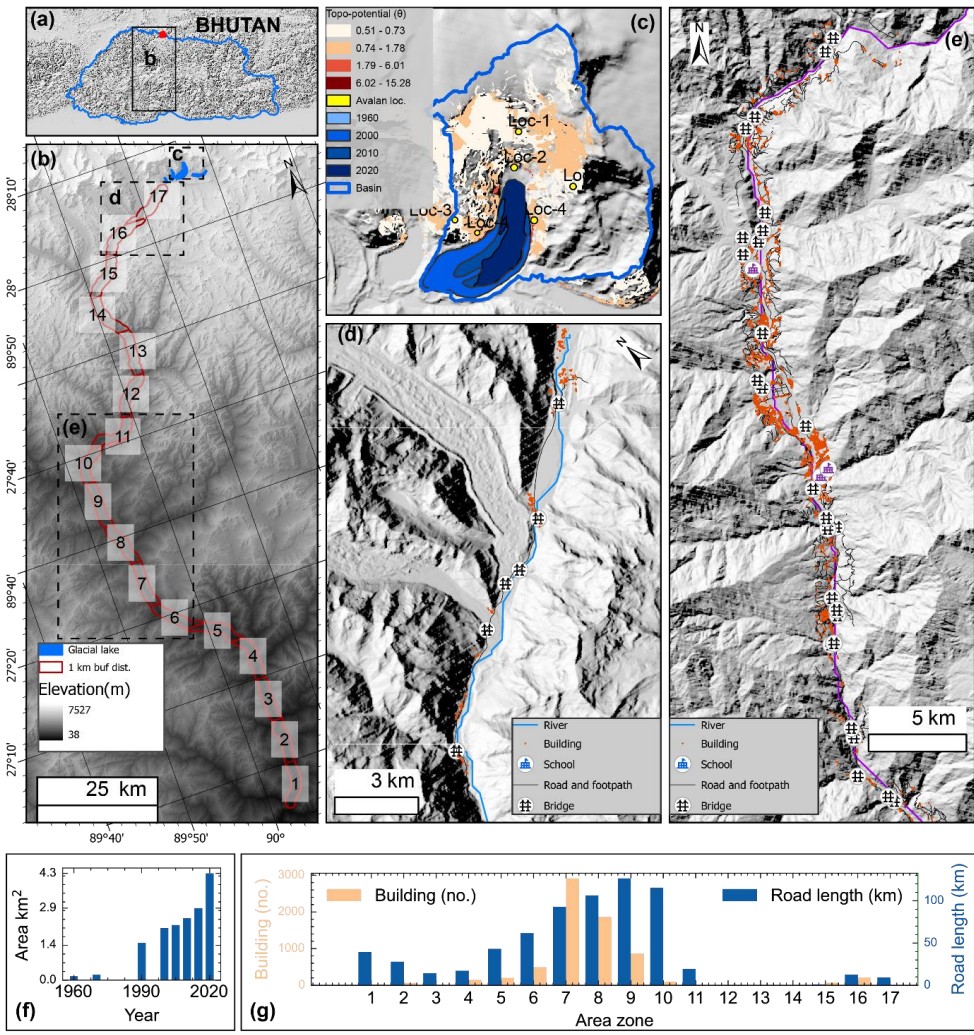


**Figure 1:** Study area. The map (a) location of Thorthormi Tsho and its downstream condition


in Bhutan. The map (b) shows elevation and the overview of glacial lakes in Lunana and


settlements along the Phochu and Punatsangchu basins, downstream of Thorthormi Tsho.


The downstream settlement is divided into 17 zones (1-17), each 10 km long. (c) Area of


Thorthormi Tsho between 1960 and 2020, and the surrounding slope with topography potential




(TPP) for mass movement entering Thorthormi Tsho. (d and e) the downstream settlements
in the (d) Lunana and (d) Punakha and Wangdue Phodrang regions. The bar graphs are (f)
the change in the area of Thorthormi Tsho between 1960 and 2020 and (g) the buildings
(count) and road (km) within the 1 km on either side of the river centreline as per the latest
OpenStreetMap.
Additionally, the Phochu and Punatsangchu basins, located downstream of Thorthormi Tsho,
are the most populated basins in Bhutan. The latest updated OpenStreetMap,
(https://www.openstreetmap.org) although it does not have 100% coverage, shows that there
are over 7000 buildings, 50 bridges, 4 schools, 687 km of road and a large area of agricultural
land within the 1 km radius of the Phochu and Punatsangchu rivers. 202 buildings are located
within the immediate 10 km downstream of Thorthormi Tsho (Fig. 1c,1d,1f). Besides, located
downstream are the two biggest hydropower plants (Punatsangchu-1 and Punatsangchu-2)
in Bhutan nearing the commission and poised to become key contributors to the nation's GDP.
Also, the Punakha Dzong, great historical and cultural significance is located downstream of
Thorthormi Tsho. This high downstream exposure to GLOF hazard further highlights the
importance of understanding GLOF characteristics from Thorthormi Tsho for GLOF modelling
(Fig. 1).
**3 Methods**
**3. 1 r.avaflow model framework**
r.avaflow is a comprehensive GIS-based open-source computational framework for modelling
mass movement from one or more release areas over the defined basal topography (Mergili
et al., 2017; Mergili and Pudasaini, 2024). It can model the entire GLOF process chain starting
from the release of avalanches, through the dynamic interaction of the avalanche and lake
water, then the overtopping and retrogressive moraine dam erosion, and finally the
downstream evolution of the resulting flow (Mergili et al., 2020b; Vilca et al., 2021; Sattar et
al., 2023). It can also robustly consider the interactions between the phases as well as erosion
and deposition (Mergili et al., 2017). Furthermore, it is equipped with a built-in function for
visualization and validation. Because of this capability, r.avaflow has been widely used to
model process chains such as GLOF in the high mountains all over the world, mostly to
reconstruct past events (Zheng et al., 2021a; Mergili et al., 2020b; Vilca et al., 2021) and to a
lesser extent to predict future hazards (Sattar et al., 2023; Allen et al., 2022).
In r.avaflow, the evolution of the flow in space and time is solved by using an enhanced version
of the  Pudasaini multiphase flow model (Pudasaini and Mergili, 2019; Pudasaini, 2012). The
flow is computed through depth-averaged conservation of mass and momentum equations for



solid and fluid components. These equations involve six differential equations accounting for
solid ($D_s$) and fluid ($D_f$) flow depths, solid ($M_{sx}$) and fluid ($M_{fx}$) momentum in x direction ($M_{sx}$ =
$D_s.v_{sx}$, $M_{fx} = D_f.v_{fx}$), and $M_{sy}$ and $M_{fy}$ in y direction ($M_{sy} = Ds.v_{sy}$, $M_{fy} = D_f.v_{fy}$), where $v$ is the
flow velocity (Mergili et al., 2017). Mohr-Coulomb plasticity is used to compute solid stress
while fluid is subjected to solid volume-fraction-gradient-enhanced non-Newtonian viscous
stress. r.avaflow also considers other factors like virtual mass force, viscous drag, and
buoyancy. These factors collectively facilitate momentum transfer between the solid and fluid
phases, enabling simultaneous deformation, separation, and mixing of phases as they
propagate across the mountain topography (Pudasaini and Mergili, 2019; Pudasaini and
Krautblatter, 2014a; Mergili et al., 2020b; Pudasaini, 2012). To numerically solve these
differential equations and propagate flow over time and space, r.avaflow uses a high-resolution
total variation diminishing non-oscillatory central differencing (TVD-NOC) scheme, a
commonly used numerical scheme to handle the advection of quantities, whilst minimising
numerical artefacts like oscillations (Wang et al., 2004). The internal friction angle and basal
friction angle, which are crucial factors governing the frictional forces influencing flow rheology,
are scaled with a solid fraction of the flow material (Mergili et al., 2018b; Mergili et al., 2017;
Pudasaini and Mergili, 2019). This scaling effectively accounts for the reduced interaction
between solid particles and the basal surface within flows rich in fluid (Mergili et al., 2018b;
Mergili et al., 2017).
r.avaflow has three different models, namely, a single-phase shallow water model with Voellmy
friction relation, an enhanced version of the multi-phase-flow of Pudasaini and Mergili (2019)
and an equilibrium-of-motion model for the slow-flow process (Mergili et al., 2017). Here, we
chose an enhanced version of the multi-phase-flow model considering an erodible moraine
dam and rock-ice avalanche as the solid component and lake water as the fluid component.
The multi-phase mass flow model can simulate the propagation of three different elements:
solid (coarse material including boulders, cobbles and gravel), fine solid (including sand and
particles larger than clay and silt), and fluid (including water and very fine particle including
clay, silt and colloids), and assign each of them with distinct flow rheology (Pudasaini and
Mergili, 2019).
Furthermore, r.avaflow has six specific optional functions including conversion of release
height to depth, diffusion control, surface control, entrainment, stopping and dynamic adaption
of friction parameters (Mergili and Pudasaini, 2024). The latest version of r.avaflow has four
options to compute erosion and entrainment, (i) calculated by multiplying the entrainment
coefficient with flow momentum, (ii) simplified entrainment-deposition numerical model of
Pudasaini and Krautblatter (2014b), (iii) a combination of (i) and (ii), and (iv) acceleration-



deceleration entrainment and deposition model. Since models (ii) to (iv) are at the
experimental phase, here, we used model (i), where the amount of entrainment is computed
dynamically by multiplying with the user-defined entrainment coefficient (CE) with the total
momentum of the flow at the given raster cell and time step (Mergili et al., 2017) (equations 1
and 2).

$$q_{E,s} = C_E|M_s+M_f|\ \alpha_{s,Emax} \quad\quad\quad\quad (1)$$

$$q_{E,f} = C_E|M_s+M_f|\ (1-\alpha_{s,Emax}) \quad\quad\quad\quad (2)$$


Where $q_{Es}$ and $q_{Ef}$ are the entrainment rates of solid and fluid respectively
$C_E$ is user user-defined entrainment coefficient ($kg^{-1}$)
$\alpha_{s,Emax}$ is using user-defined solid entertainable material height (m)

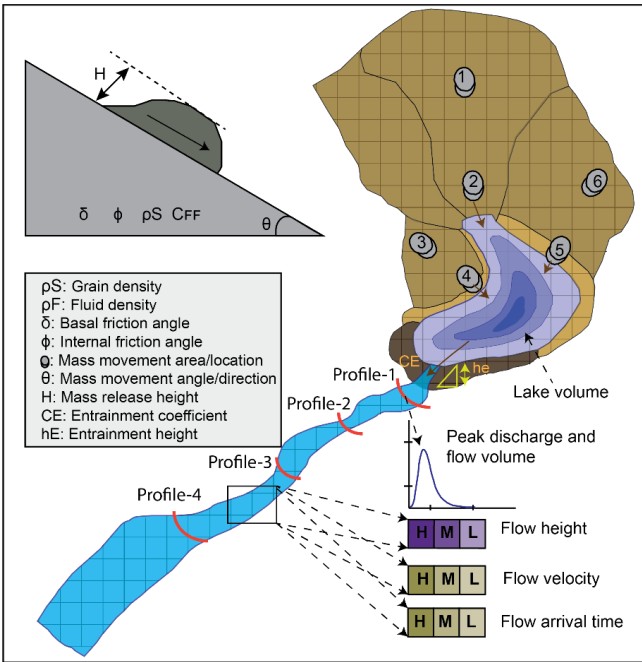


**Figure 2:** Schematic view of Thorthormi Tsho, surrounding terrain (study area) and input
parameters employed for investigating r.avaflow model output sensitivity used in this study. 1-
6 shows the location of the mass movement areas into the lake.



We utilized r.avaflow direct (Mergili and Pudasaini, 2024), a web-based tool, to initially
generate the sample model script. We modified it by inputting parameters relevant to each
experimental set-up and wrote a bash shell script for all simulations in each experiment to test
various parameter values within our predefined range. We developed one master bash script
for each experiment that allowed us to run all experiments in parallel leveraging the Rocket
High Performance Computing (HPC) facilities at Newcastle University. All the GLOF
simulations were done for Thorthormi Tsho and were run for 1500 seconds when the flow
reaches up to ~24 km downstream of the lake depending on values of various parameters
defined here. The flow propagation beyond this point and its interaction with the downstream
component are beyond the scope of this study.
**Table 2:** Key parameters tested in this study to investigate model output sensitivity.  Detailed
parameters for r.avaflow modelling are provided in Table S1.

| Parameter | Value range | No. of simulations | Constant value |
|---|---|---|---|
| Topographic data (DEM) and Mesh size | High Mountain Asia DEM (HMA-DEM) (8m), AW3D30 (30m), NASADEM (30m), SRTM GL3 (90 m) | 12 (3×4) | HMA-DEM |
| Avalanche origin location | Left (2), Right (2), Headwall (2) | 6 | Loc-1 |
| Avalanche volume | $1 - 10 \times 10^6$ m$^3$ | 10 | $5 \times 10^6$ m$^3$ |
| Grain density | 1000 – 2700 kg/m$^3$ | 10 | 2700 |
| Entrainment coefficient | -5.85 – -6.95 kg$^{-1}$ | 10 | -6.35 |
| Basal friction angle | 10 – 14$^\circ$ | 10 | 10 |
| Internal friction angle | 25 – 35$^\circ$ | 10 | 28 |
| Fluid friction number | 0.027 – 0.050 | 10 | 0.05 |

**3.2 Model inputs parameterisation and experimental setups**
r.avaflow has a large choice of parameters and initial conditions, such as a DEM representing
initial basal topography, the volume of the solid and liquid phase, entrainment and stopping
parameters and density and friction parameters (Mergili and Pudasaini, 2024) (Table S1). The
values specified for these parameters influence crucial aspects of modelled GLOF flow,
including impacted area, travel distance, travel time, and volume of sediment deposited at the



various downstream locations (Mergili et al., 2017). In this study we selected a total of nine
parameters which are identified as important in the previous studies (e.g. Mergili et al.
(2020a)): 1) DEM dataset, 2) mesh size; 3) the origin of mass movement into the lake; 4)
volume of mass movement entering the lake; 5) grain density of mass movement entering the
lake; 6) entrainment coefficient; 7) basal friction angle; 8) internal friction angle; 9) fluid
frictional number. To investigate the impact of DEM dataset variation (1) and mesh size
variations (2), we modelled GLOF by employing freely available global and regional DEM
datasets with differing spatial resolution and vertical accuracy (Table 2). For the origin of the
mass movement entering lake (3), we first computed the topographic potential for slope
movement into the lake (Allen et al., 2019) (Fig. 1B) and selected six different sites by
considering the topographic potential values and direction of the lake (Fig. 2). The volume of
mass movement entering lake (4) was varied between $1 \times 10^6$ $m^3$ and $10 \times 10^6$ $m^3$. The
avalanche grain density ($\rho S$) (5) value range was considered based on assumed combinations
of rock and ice avalanche parts following the approach used in the earlier studies (Allen et al.,
2022; Sattar et al., 2023). For parameters 6-9, we gathered various values employed in
previous studies (Allen et al., 2022; Mergili et al., 2020a; Mergili et al., 2020b; Vilca et al.,
2021) and established the conservative range. In doing so, we computed descriptive statistics
and established the median, upper quantile value, and lower quantile for each parameter using
these collated values (Fig. S1). We then varied these parameter values between the
calculated upper quartile and lower quartile, to give 10 equally spaced values in total. This
range of 10 values was utilised in our 10 experiments for the respective parameter, whilst
holding other parameter values constant at the median value. For example, for the internal
friction angle ($\phi$) experiment, the $\phi$ was varied between the upper and lower quantiles, with 10
increments in total, whilst holding constant the other parameter values (Table 1).  An overview
of employed parameters and workflow is shown in Fig. 2 and Table 1, while further details on
the parameter range used for each experiment are provided in the following section.








**Table 2:** Characteristics of DEM datasets employed in this study to investigate the impact of
DEM dataset variation on GLOF modelling results.

| DEM dataset | Acquisition techniques | Spatial resolution | Vertical accuracy | Coverage | Survey date |
|---|---|---|---|---|---|
| AW3D30 | Optical stereo images | ~30 m | 6.84 m (RMSE relative to ICESat in HMA) (Liu et al., 2019) | Global | 2006 to 2011 |
| NASADEM | SAR Interferometry | ~30 m | 5.3 m (RMSE for USA) (Liu et al., 2019) | Global | 2000 |
| SRTM GL3 | SAR Interferometry | ~90 m | 9.51 m (RMSE relative to ICESat in HMA) (Buckley et al., 2020) | Global | 2000 |
| HMA 8m DEM | Optical stereo images | 8 m | 2-m (depending on the type of sensor) (Shean, 2017a) | High Mountain Asia (HMA) | 2002 to 2016 |

### 3.2.1 Digital elevation (1) model and mesh size (2)

Here our goal is to constrain model output uncertainty stemming from the use of freely
available global and regional DEM datasets. We performed a series of GLOF simulations
using four open-access DEM data of various resolutions, vertical accuracy and elevation
derivation methods, namely, High Mountain Asia DEM (HMA-DEM; 8 m) (Shean, 2017b),
ALOS Global Digital Surface Model (AW3D30; 30 m) (Jaxa, 2021), NASADEM (~30 m) (Nasa-
Jpl, 2021), and SRTM GL3 (~90 m) ((Srtm), 2013). Further to investigate the impact of mesh
size variation in each DEM dataset, we performed three simulations for each DEM data by
changing mesh size to 20 m, 30 m, and 40 m. The GLOF simulations for all other parameter
experiments were done using HMA-DEM at 8 m resolution (Table 2).

### 3.2.2 Volume of lake and avalanche entering lake (4)

r.avaflow has the option to define the initial release volume of different phases involved in the
GLOF process chain. Here, we assume GLOF was initiated by rock-ice mixed mass
movement entering into the lake followed by a tsunami wave hitting the moraine damming the
lake and causing moraine dam failure. Accordingly, we defined the frontal moraine damming
Thorthormi Tsho as phase-1 (rock component with $\rho$ = 2700 kg m$^3$), mass movement entering
Thorthormi Tsho as phase-2 (rock-ice component) and Thorthormi Tsho as phase-3 (fluid part).
Conducting a bathymetry survey of Thorthormi Tsho is highly challenging as the lake is filled
with debris and icebergs. Therefore, we considered the volume by considering the mean value
(294×10$^6$ m$^3$) of all the volumes estimated from a total of eight area-volume scaling equations
(Table S2). This same calculated volume is used as constant fluid volume across all the GLOF
simulation experiments we conducted here and was not considered for sensitivity analysis.



However, r.avaflow requires spatially varying lake bathymetry to be used as fluid release
height rather than the absolute value of lake volume. Fortunately, Thorthormi being a recently
formed lake, has ice thickness data covering the extent of the lake (Farinotti et al., 2019).
Therefore, we computed the bathymetry of Thorthormi Tsho by subtracting ice thickness data
from the surface DEM (Linsbauer et al., 2016; Linsbauer et al., 2017). Assuming that the
present-day lake has been formed by filling the over-deepening, this ice-thickness-derived
bathymetry was adjusted to match the volume we calculated from the empirical equation
(Table S2).
The volume of the avalanche entering the lake serves as a fundamental parameter for defining
various scenarios in the forward modelling of a GLOF (for example, Allen et al. (2022) and
Sattar et al. (2023)). However, for the forward modelling purpose, it is difficult to predict how
big or small the avalanche will be. Considering these uncertainties, to test the effect of mass
movement of various volumes, we conducted a series of 10 experiments considering volumes
ranging from $1 \times 10^6$ to $10 \times 10^6$ m$^3$ (Table 1).

### 346    3.2.3 Origin of mass movement into the lake

To account for uncertainties in the exact origin of mass movement into the lake, we identified
a total of six mass movement areas, each characterised by different directions, distances, and
angles to the lake (Fig. 1 and Fig. 2). To do this, we first computed topographic potential for
ice/rock avalanche and landslide movement into the lake based on slope and run-out trajectory
criteria (Allen et al., 2019). Based on this first-order estimate, we identified the six potential
avalanche source areas: Loc-1 (slope at ~900 m away from the headwall), Loc-2 (headwall),
Loc-3 (slope at the ~900 m from right moraine dam), Loc-4 (right moraine dam), Loc-5 (slope
at ~900 m from left moraine dam), Loc-6 (left moraine dam) (Fig. 1 and Fig. 2).  We then ran
one scenario for each potential avalanche input location we identified.

### 356    3.2.4 Grain density of mass movement entering lake (5)

Our goal here is to assess the impact of the grain density of the mass movement entering the
lake, which serves as a proxy for the ice-to-rock ratio. Accordingly, we consistently set the
grain density of phase-1 (moraine) at 2700 kg m$^3$ across all the experiments, whilst the fluid
density of phase-3 was also held constant at 1000 kg m$^3$. In the earlier studies, the grain
density of mass movement entering the lake has been used as a proxy of the portion of an
ice-rock component of mass movement into the lake, which is highly uncertain (Vilca et al.,
2021; Allen et al., 2022). The phase separation of rock and ice components of the mass
movement with different densities is not well established in  r.avaflow (Vilca et al., 2021).
Therefore, in this study, following Sattar et al. (2023), a portion of snow and ice in the





avalanche is considered fluid by adjusting the material density of the phase-2 represented by
the avalanche (Table S3). In our experiment set-up, this is executed by varying the density
value between 2700 kg m$^{-3}$ (representing 100% rock) to 1000 kg m$^{-3}$ (representing 100%
water) (Table 1).

**3.2.5 Entrainment coefficient (6)**

Material entrainment due to bed erosion can make the flow more concentrated and thus
increase the volume, resulting in spatial and temporal variation of flow. In the r.avaflow model,
the user must define entrainment height in the form of a raster covering the entire model
domain, which can be either identified using remote sensing imagery or fieldwork (Mergili and
Pudasaini, 2024). However, here, we considered frontal moraine damming the lake as the only
entrainment height (Fig. 1). The amount of entrainment itself is dependent on the user-defined
entrainment coefficient ($C_E$). In r.avaflow the logarithm with base 10 of the $C_E$ must be entered
(Mergili et al., 2018a; Mergili et al., 2017). Here, we modelled 10 scenarios of GLOF by varying
$C_E$ between $10^{-6.95}$ to $10^{-5.85}$ kg$^{-1}$ (Table 1).

**3.2.6 Frictional parameters (7-9)**

The internal friction angle ($\phi$), basal friction angle ($\delta$) and fluid friction number ($C_{FF}$)
mechanically control the basal shear stress, internal deformation, anisotropy of the stresses,
and hydraulic pressure gradient of the solid constituents (Pudasaini and Krautblatter, 2014a),
which are essential attributes influencing flow runout distance and time. Within the r.avaflow
model set-up, a user can either use spatially varying values for these frictional parameters
using a raster map or one absolute value (Mergili and Pudasaini, 2024; Mergili et al., 2017).
In this study, we computed 10 experiments for each of these frictional parameters. Specifically,
by varying the $\phi$ between 25° to 35°, $\delta$ between 10° and 14° and $C_{FF}$ between 0.027 to 0.050
(Table 1).

**3.3 Sensitivity Analysis**

Here we use sensitivity analysis, to determine how variations in the initial values for key impact
the model outputs (Saltelli et al., 2004). Thus, our goal is not to determine the 'correct' value
for each parameter but to determine the r.avaflow input parameter(s) that cause the most
variation in the model output. To constrain this variability, we mainly focused on examining the
peak discharge, total discharge, and flow arrival time as the output metrics. The flow for all the
experiments was measured from the profile immediately beneath the moraine dam (profile-1
in Fig. 2). We calculated the peak and total discharge based on the flow data obtained from
the same profile (Fig. S2). The flow arrival time was considered as the average value across





the time recorded from the profiles located 3 km, 6 km and 9 km downstream of the Thorthormi
Tsho (profile-2, 3, 4 in Fig. 2). All input parameters were standardized within a percentile range
of 0 to 100 for comparative analysis of their effects on the resultant outputs. For the scalable
parameters, we also computed simple linear regression considering input parameters as the
independent variable and model output as the dependent variable. To ascertain the sensitivity
of the model output to variations in value across all parameters, we computed the coefficient
of variation (CV) for individual parameters and subsequently ranked them based on this metric.
The CV is a statistical measurement of the dispersion of data points around the mean,
regardless of the units used to measure it. CV is deemed suitable here since the r.avaflow
output variability is caused by input parameters that are measured in different units. To
calculate CV, we took the standard deviation of the output value range of a particular
experiment (e.g. peak discharge) and divided it by the mean of the same output range (Abdi,

411    2010).

**4 Results**
**4.1 Effect of DEM dataset**
When the GLOF is modelled employing freely available global and regional DEM datasets
(HMA-DEM, AW3D30, NASADEM, SRTM GL3), our results showed a variation of peak and
total discharge of GLOF from the Thorthormi Tsho by almost 100% and 400%, respectively
(Fig. 3). Specifically, HMA-DEM consistently produced the lowest GLOF magnitude, while
SRTM GL3 consistently produced the highest. The peak flow fluctuates between 10-115% and
the total discharge between 55-400% (Fig. 3). Although NASADEM and AW3D30 have a
similar spatial resolution, notable differences (65%) in peak discharge emerged between
simulations done using these datasets (Fig. 3b and 3c).
We observed a significant fluctuation in the mean flow height (82%) and velocity (65%) along
the flow path resulting from the change in DEM datasets (Fig. 3). For instance, the mean flow
height along the river centreline ranged from 39 m (HMA-DEM) to 54 m (SRTM GL3) (Table
3) and the flow reach distance increased from 15.5 km (HMA-DEM) to 24.2 km (SRTM GL3).
Once again, NASADEM and AW3D30 yielded significantly different maximum flow heights
(8.5%) and reach distances (72%) (Fig. 3b and 3c). The use of various sources of DEM
datasets led to variations in total flow arrival time by around 16%. Flows derived from SRTM
data always arrived earlier, while those using HMA-DEM consistently showed the latest arrival
times (Table 3). For example, at 5 km downstream, SRTM GL3 showed the earliest arrival at
3.46 min while HMA-DEM resulted in the latest arrival at 4.37. The portion of the solid





component of the flow did not exhibit significant fluctuations in response to changes in input DEM datasets (Fig. 3).

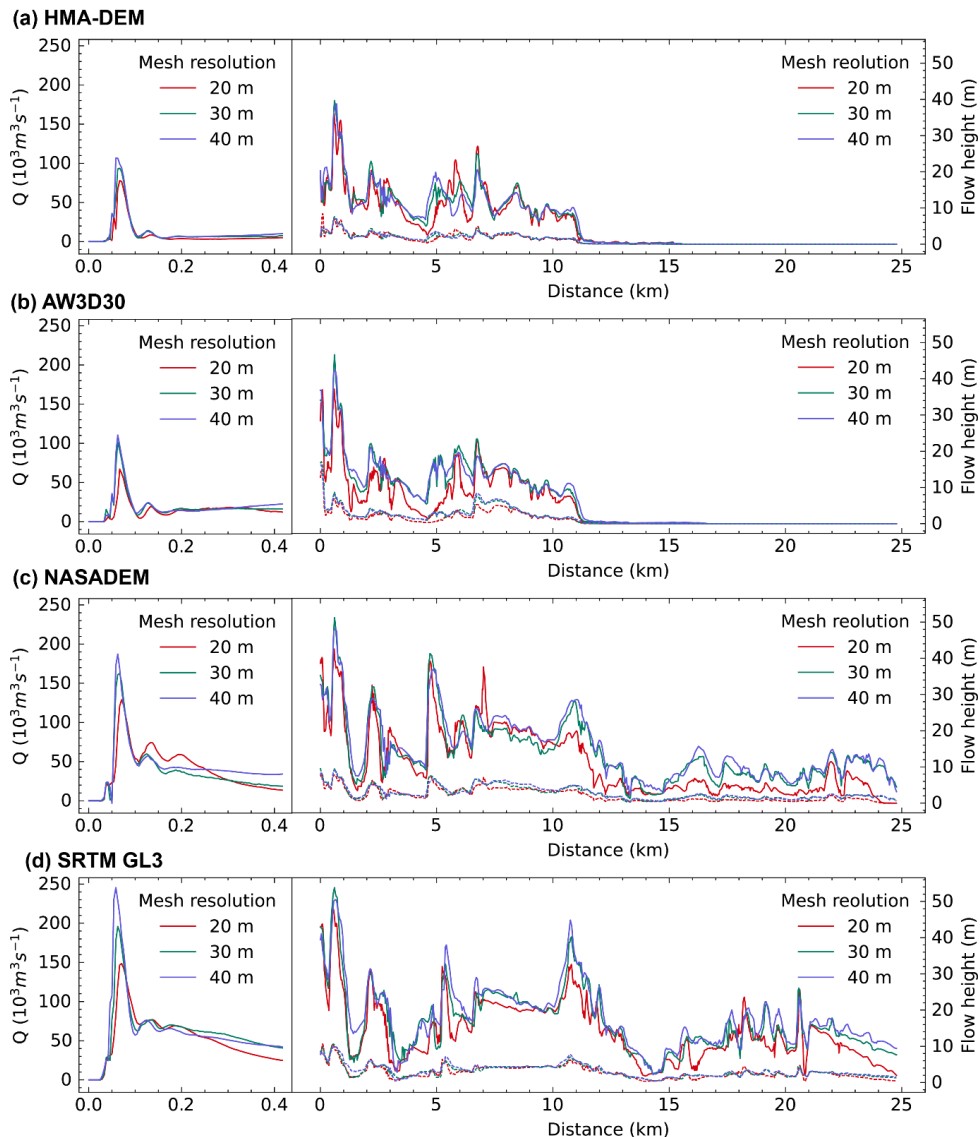

**Figure 3:** Hydrographs (right panels) and maximum flow height along the river centerline (left panels) generated by conducting a sequence of r.avaflow simulations, employing different types of DEM datasets and varying the mesh resolution.

**4.2 Effect of mesh size variations**



When mesh size was increased from 20 m to 30 m and 40 m across all the DEM datasets,
we noted a substantial increase in peak and total discharge, although changes in resulting
flow characteristics like flow velocity were minimal (Fig. 3). For instance, in the case of the
experiment with HMA-DEM, the peak discharges increased to 20% and 38%, respectively
(Fig. 3). However, the mean flow velocity increased only by 6% when the mesh size was
increased from 20 to 40 m (Table 3). Likewise, there was no significant difference in the flow
reach distance emerging from changing mesh size (Fig. 3). For instance, flow with all three
mesh sizes for HMA-DEM resulted in to flow reach distance of about 15 km (Fig. 3a). Mesh
size variation resulted in arrival flow time variation of about 20%, with 40 m leading to earliest
arrival and 20 m the latest (Table 3).
**Table 3.** Percentage change in flow velocity, depth and arrival time resulting from variation in
values of different input parameters we employed in this study. The total percentage (%)
change represents the output variation between the maximum and minimum values used in
the experiment. The average percentage (%) change is calculated as the mean change across
all incremental steps employed in setting up the experiment. The arrival time average of the
record from three locations, Profile-2, -3, and -4) (Fig. 2). Flow velocity and depth are mean
values taken from the river centreline. The detail flow pattern is provided in Fig. S3, Fig. S4
and Fig. S5.

| SL no. | Parameter | Velocity (% change) | | Depth (% change) | | Time (% change) | |
|---|---|---|---|---|---|---|---|
| | | Average | Total | Average | Total | Average | Total |
| 1 | DEM dataset | 16.25 | 65 | 20.5 | 82 | 4 | 16 |
| 2 | Mesh Resolution | 2 | 6 | 3 | 9 | 4 | 12 |
| 3 | Volume of mass movement entering lake | 9.2 | 92 | 92.3 | 923 | -14.3 | -143 |
| 4 | Density of mass movement entering lake | 0.2 | 2 | 3.1 | 31 | 6 | 6 |
| 5 | Location of origin of mass movement entering lake | 3.7 | 37 | 8.2 | 82 | 8 | 8 |
| 6 | Entrainment coefficient | 1 | 10 | 4.9 | 49 | 3 | 3 |
| 7 | Basal friction angle | 2.3 | 23 | 4.2 | 42 | 6.8 | 68 |
| 8 | Internal friction angle | 0.1 | 1 | 3.8 | 38 | 0 | 0 |
| 9 | Fluid friction number | 5.5 | 55 | 7 | 70 | 0.8 | 8 |



### 4.3 Effect of origin of mass movement entering the lake

Our study found that the GLOF process chain initiated by mass movements from various locations (Loc-1 to Loc-6) results in a significant fluctuation in the GLOF output (Fig. 4). The peak discharge varied by approximately 200% and the total discharge by 55% (Fig. 4). Likewise, the mean flow height and velocity also fluctuated by 65% and 82%, respectively (Table 3). By comparison, the flow resulting from the GLOF initiated by mass entering from the Loc-1 (Fig. 4a) (900 m from the headwall) and Loc-5 (Fig. 4e) produced the highest magnitude GLOF and that from the loc-4 (Fig. 4d) was the lowest.  For example, the highest peak (18 ×10$^3$ m³) and total discharge (11 ×10$^6$ m³) occurred from Loc-1, while the lowest peak (6,000 m³) and total discharge (7 × 10$^6$ m³) were from Loc-4 (right lateral moraine) (Fig. 4a and 4d). The longest flow reach distance (25 km) was produced by loc-1 and loc-5, while the shortest was from minimum from loc-3 (10 km) (Fig. 4c).  Arrival times vary approximately by 20%, where the flow from Loc-5 arrives earlier while Loc-1 arrives at the latest (Table 3 and Fig. 4). Solid volumetric portion did not exhibit significant fluctuation, with concentration ranging from 4% (Loc-4) to 5% (Loc-2) (Fig. 4).



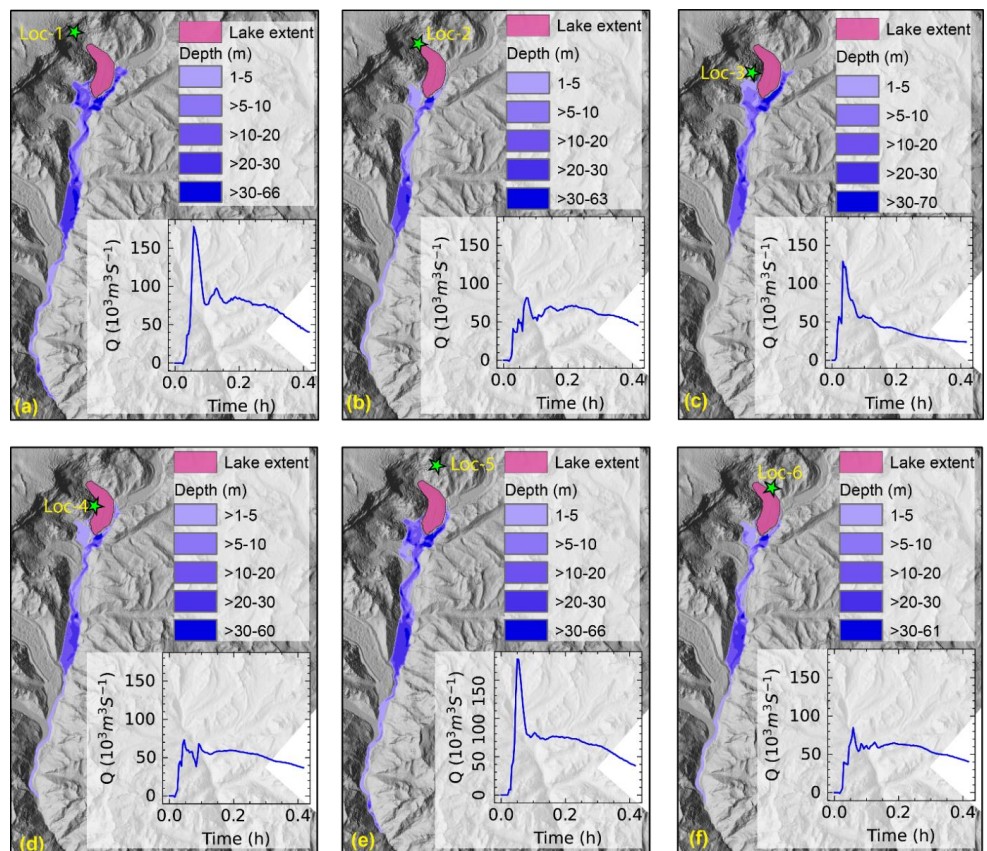

**Figure 4:** Flow rate and depth resulting from mass movement into the lake from different locations: loc-1 (a) to loc-6 (f).

## 4.4 Effect of volume and grain density of mass movement entering the lake

To separate the effect of variation in volume and density (ρS) of mass movement entering the lake, we simulated all 10 scenarios of the GLOF event using the mass movement initiated from loc-1. Here we observed that only volume variation in mass movement leads to a very large variation in the resulting peak (1160%) (Fig. 5a) and total flow (2500%) (Fig. 6a). Subsequently, this resulted in maximum variation in flow characteristics such as mean flow height (923%) and flow arrival time (50%) (Table 3, Fig. S3, Fig S5). Conversely, the ρS variation showed the least impact on both peak (5%) and total discharge (24%) (Fig. 5b, Fig. 6b, and Fig. 7b) and subsequent characteristics such as flow height (3%) and velocity (2%) (Table 3 and Fig. S5). Both volume and density variation did not result in significant fluctuation in the solid-volumetric concentration of the flow (Fig. S3).

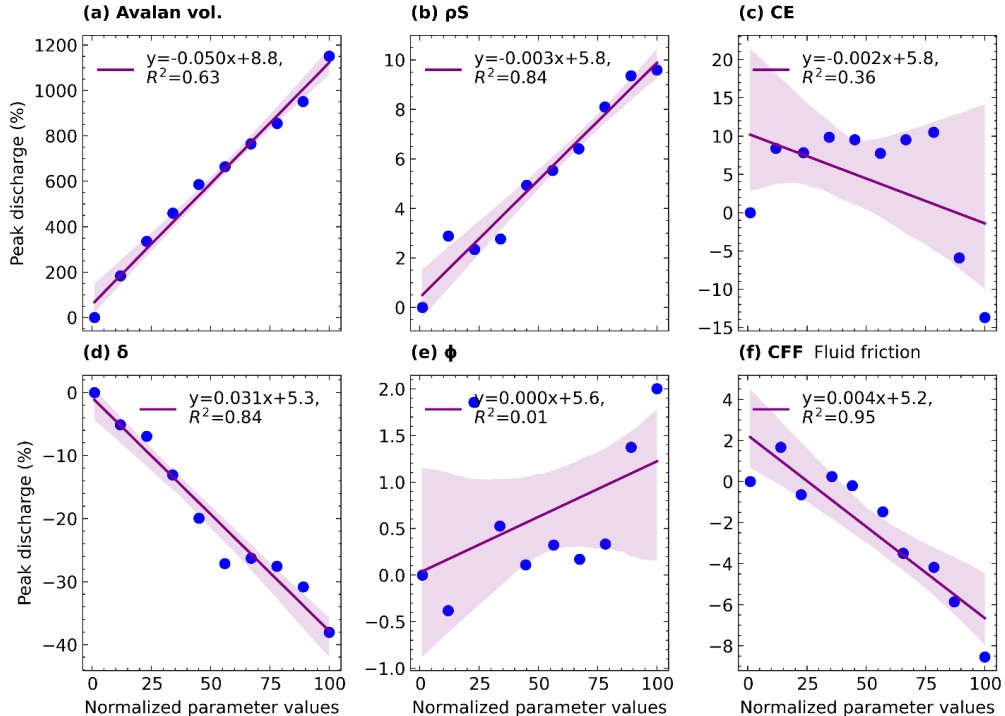

**Figure 5:** Linear regression between input parameter value variation and resulting peak discharge. All input parameter values are normalized between 0 to 100. The linear regression is computed only for the volume of mass movement into the lake (a), grain density (b), entrainment coefficient (c), basal friction angle (d), internal friction angle (e) and fluid friction number (f).

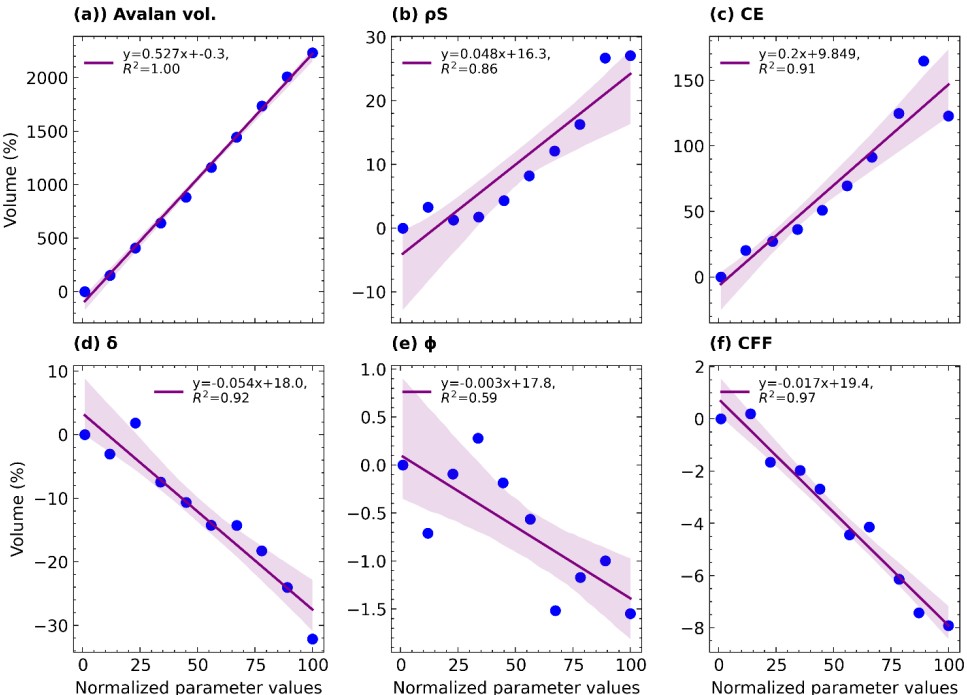

**Figure 6:** Linear regression between input parameter value variation and resulting total discharge. All input parameter values are normalized between 0 to 100. The linear regression is computed only for the volume of mass movement into the lake (a), grain density (b), entrainment coefficient (c), basal friction angle (d), internal friction angle (e) and fluid friction number (f).




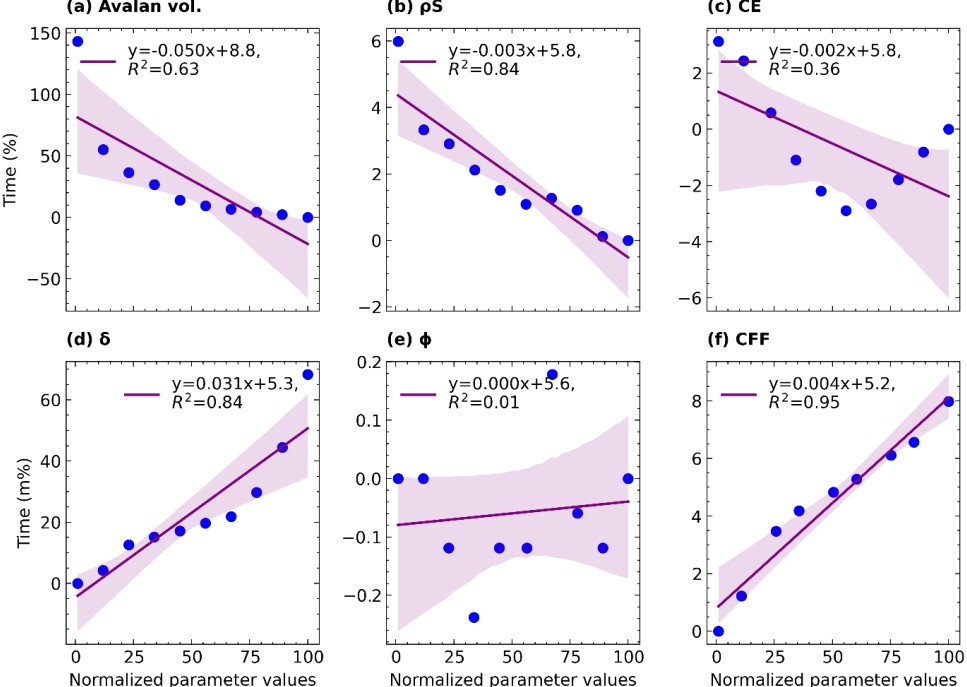

**Figure 7:** Linear regression between input parameter value variation and flow arrival time. All input parameter values are normalized between 0 to 100. The linear regression is computed only for the volume of mass movement into the lake (a), grain density (b), entrainment coefficient (c), basal friction angle (d), internal friction angle (e) and fluid friction number (f)

**4.5 Effect of entrainment coefficient**

Variations in the entrainment coefficient substantially impact the resulting GLOF output, causing fluctuations in peak discharge and volume by 13% and 123%, respectively (Fig. 5c and 6c). These changes also affect the flow characteristics including mean depth (49%) and reach distance (20%) (Table 3) but had minimal effect on arrival time (3%) (Fig. 7c). Most notably, unlike other parameters, entrainment variation also affected the solid concentration of the flow (Fig. S3). An increase in the entrainment coefficient from $10^{-6.95}$ to $10^{-5.85}$ kg$^{-1}$ led to a 30% increase in the mean solid volumetric concentration of the flow.

**4.6 Effect of frictional parameters**

Among the frictional parameters, the variation in basal friction angle (δ) resulted in a significant fluctuation in GLOF magnitude and resulting flow characteristics (Fig. 5d, 6d and 7d). While the variation of fluid friction angle had minimal impact on the resulting peak and total flow (Fig. 5e, 6e), it notably altered other flow characteristics, such as flow velocity (55%) and depth





(70%) (Table 3). The δ angle increase from 10 to 14° resulted in a peak and total discharge
decrease of 36% (Fig. 5d) and 32% (Fig. 6d), respectively. Likewise, the flow velocity
decreased by 23% resulting into delay in flow arrival by 18% (Table 3). Conversely, the peak
flow decreased by 2% only in response to an increase in the internal frictional angle from 25-
35° (Fig. 5f). The variation in all frictional parameter values did not result in a significant change
in the solid volumetric concentration of the flow (Fig. S4).

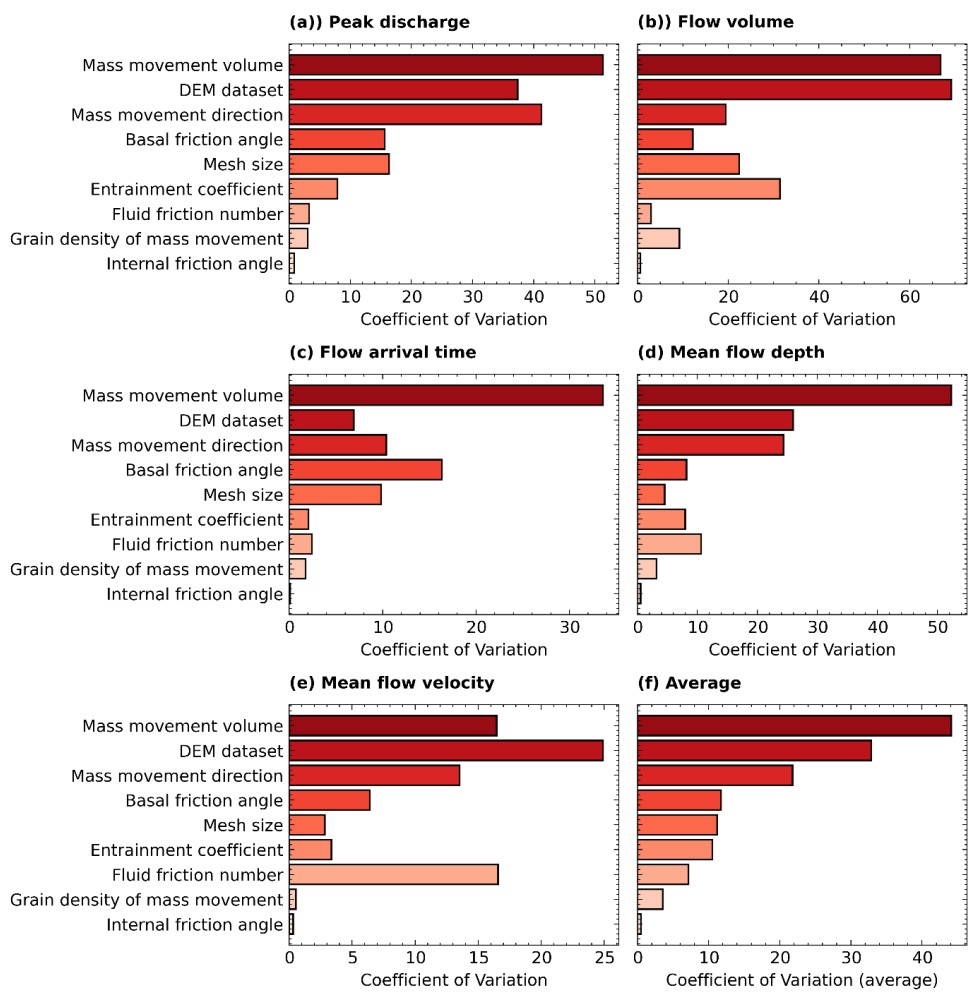


**Figure 8:** The coefficient of variation for (a) peak flow, (b)volume, (c) time, (d) average flow
height along the river centreline, (e) flow velocity along the river centreline and (f) average
across all these output parameters.



### 4.7 Comparison of the effect of all parameters


To compare output sensitivity resulting from all parameters and initial conditions considered
here, we calculated the coefficient of variation (CV) for peak flow, total discharge, arrival time,
flow height and flow velocity. We further computed the average coefficient of variation (avg.
CV) across all these output variables and examined the overall impact of each input parameter
variation. Comparing all these output indicators, mass movement entering the lake had the
greatest impact (avg. CV = 47%), followed by DEM datasets (avg. CV = 35%) and the origin
of mass movement (avg. CV = 21%). Other input parameters like mesh size, basal friction
angle ($\delta$), and entrainment coefficient also caused significant variation in resulting GLOF.
Notably, fluid friction number had a significant impact on flow height with its CV = 16% despite
having minimal impact on other flow characteristics.
For the six scalable parameters, we computed linear regression (Fig. 5 to Fig. 7). The linear
regression analysis unveiled that the four parameters, namely volume ($R^2$=0.99) of mass
movement into lake, $\rho S$ of mass movement into lake ($R^2$=0.96), basal friction angle ($\delta$) ($R^2$ =
0.96) and $C_{FF}$ ($R^2$ = 0.83) offer strong explanatory power regarding the variability observed in
resulting GLOF peak discharge (Fig. 5). Among these sets of parameters, volume (m = 1.6)
and $\rho S$ (m = 0.085) of mass movement entering lake indicated a positive relationship while $\delta$
(m =-0.347) and $C_E$ (m=-0.091) exhibited a negative relationship (Fig. 5). By contrast, the
internal friction angle ($R^2$ = 0.24) and entrainment coefficient ($C_E$) ($R^2$ = 0.22) exhibited a weak
relationship with the peak discharge. All six parameters ($R^2$ >0.9) except for the internal friction
angle ($R^2$ =0.59) indicated a high level of explanatory power regarding the variation of resulting
total discharge. Across all six parameters, the volume of the avalanche exhibited the highest
$R^2$ value, signifying a strong explanatory power regarding the modelled discharge volume
compared to the other parameters. Additionally, the substantial magnitude of the slope (m=1.6
and m=0.53 for peak and total discharge, respectively) associated with the volume of
avalanche further underscores its high magnitude relationship with the modelled GLOF flow,
surpassing that of the other parameters (Fig. 5a and 6a).
Basal friction angle $\delta$ and $C_{FF}$ demonstrated a high level of explanatory power concerning the
variability in flow arrival time, as evidenced by their $R^2$ values of 0.98 and 0.97, respectively.
Avalanche volume variation also exhibited a high explanatory power with a negative
relationship, supported by an $R^2$ of 0.81 and a slope (m) of -0.019, although the linearity
became less pronounced within the volume range of $4 \times 10^6$ m$^3$ to $10 \times 10^6$ m$^3$. In contrast,
other parameters, including $C_E$, and $\phi$, did not exhibit a definitive linear relationship. (Fig. 7).
However, CE variation showed a threshold effect on arrival time; increasing the CE from 10$^-$





$^{5.95}$ to $10^{-6.42}$ kg$^{-1}$ decreased arrival time, while further increases towards $10^{-5.85}$ kg$^{-1}$ led to a
linear increase in arrival time.

**5 Discussion**

Our primary aim was to investigate the sensitivity of the model GLOF outputs from r.avaflow
to a range of values for key model input parameters. Previous studies have underscored the
sensitivity of r.avaflow model outputs to various input parameters, including basal friction
angle, entrainment coefficient and volume of avalanche entering the lake (Baggio et al., 2021;
Mergili et al., 2018b; Mergili et al., 2020a). This study advances our understanding of GLOF
modelling by conducting a comprehensive sensitivity analysis across nine parameters and
multiple GLOF simulations. As a result, we have for the first time, ranked these nine GLOF
input parameters based on their contributions to model output variabilities. Our results showed
that modelled GLOF output parameters are substantially sensitive to six of the nine
parameters we tested here (DEM dataset, mesh size, volume of mass movement into the lake,
origin of mass movement into the lake, entrainment coefficient, and basal friction angle)
suggesting that GLOF modelling results are subject to uncertainty from the multiple sources.
The findings offer valuable perspectives on the uncertainty of GLOF modelling results and
complexities inherent in modelling the GLOF process chain within the rugged mountain terrain
such as in the Himalaya.

**5.1 DEM datasets and mesh size variations**

DEM is one essential data for GLOF and other flood modelling (Hawker et al., 2018; Saksena
and Merwade, 2015; Schumann and Bates, 2018; Westoby et al., 2014). The impact of DEM
resolution is even more pronounced in the steep and complex topographic conditions
prevalent in high mountain regions like the Himalaya (Liu et al., 2019). Our study provides the
quantification of the effect of DEM in such environments for the first time. Our results suggest
that the use of global and regional DEM datasets ranging from HMA-DEM (8 m) to SRTM GL3
(90 m) leads to over two-fold and four-fold variations in peak and total discharge, respectively,
and cause successive significant fluctuations in flood height, reach distance and flow arrival
time. This likely results from the low-resolution DEMs not fully resolving the river channel
compared to higher resolution DEM, leading to reduced river channel conveyance (Fig. 9 and
Fig. S7) (Muthusamy et al., 2021). This was supported by a comparison of the DEM profile
and flow height along the river centreline (Fig. 9) and across the multiple vertical cross-
sections along the river channel (Fig. S6). The analysis showed that GLOF output from SRTM
GL3, where river channels are poorly resolved, was comparatively higher than that from the
HMA-DEM with the better resolved channel. Also, DEM datasets were acquired at different





times, meaning the topographic features they captured might also differ depending on natural
geomorphological change or human-made alteration of the earth's surface over time
(Schumann and Bates, 2018).

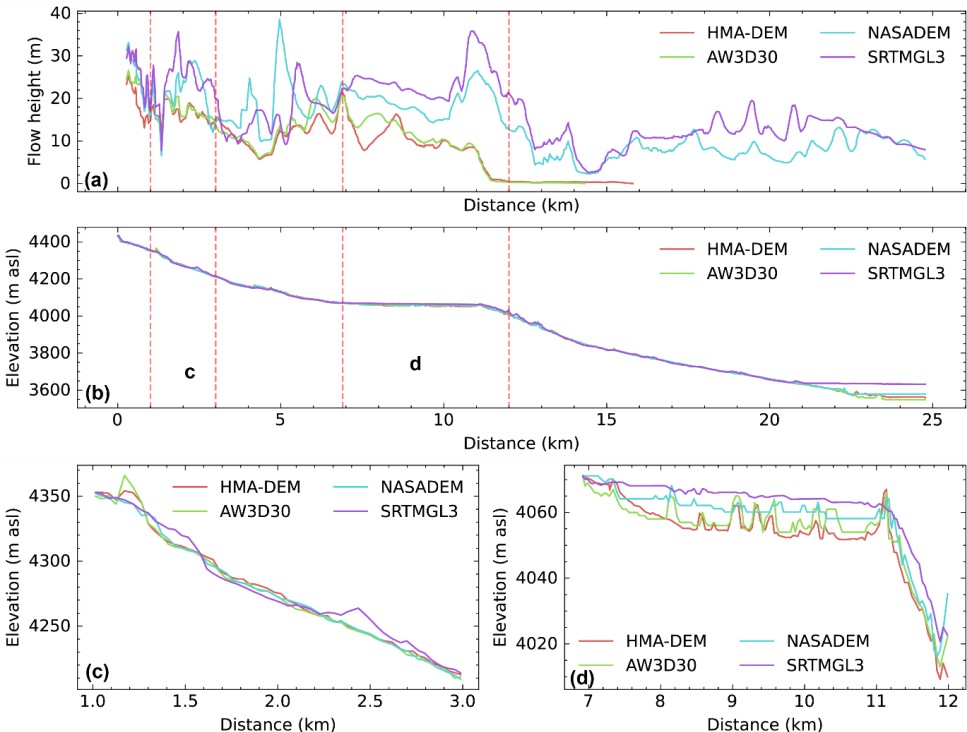


**Figure 9:** A comparison of the elevation profiles from four DEM datasets and the
corresponding flow depths along the river centreline, generated through r.avaflow modelling.
panels (a) and (b) show the flow depths and elevation profiles along the river centreline.
Panels (c) and (d) illustrate elevation profiles for two specific sections. DEM and flow height
profiles from the vertical cross sections at various distances are also provided in the
supplementary figure (Fig. S6). The DEM datasets were co-registered using Shean et al.
605  (2016).

Overestimation of flood maps stemming from reductions in DEM resolution has been reported
in urban flood modelling (Muthusamy et al., 2021; Mcclean et al., 2020). However, the impact
of DEM data on GLOF modelling, especially in a complex topographic setting such as in the
Himalaya has been rarely documented (Wang et al., 2011). Our results show the substantial
variation in GLOF model output stemming from DEM dataset variation, even when employing
DEM with comparable spatial resolutions, which underscores the criticality of high-quality DEM
data in GLOF modelling (Fig. 9). DEM datasets covering rugged high mountain terrain, where





GLOFs typically occur are likely to have more errors due to geometric distortion and data loss
due to challenges involved in data acquisition for DEM production (Hugonnet et al., 2021; Liu
et al., 2019). Therefore, using global scale DEMs, such as SRTM and ASTER, for GLOF
modelling due to the absence of high-resolution alternatives (Wang et al., 2011) may only be
suitable for first-order assessment of GLOFs at large scales  (Zhang et al., 2023b). This is
important as uncertainty stemming from DEM datasets is often overlooked and/or not well
addressed in the previous basin-specific GLOF modelling work (Rinzin et al., 2023; Sattar et
al., 2023; Sattar et al., 2021b).

### 5.2 Mass movement origin variation

Our study indicated that different locations of avalanche initiation produced GLOFs with
approximately two-fold variations in their peak discharge, volume, and reach distance (Fig. 4).
These variations can be explained based on the lake geometry and the direction/ angle at
which the mass movement enters the lake. r.avaflow model provides detailed output
parameters such as kinetic energy associated with the flow, and flow height map for each time
step, which allowed us to better understand the cause of this variation. For example, the
avalanche mass originating from loc-1, which is located at the slope above the headwall,
directly impacts the head end of the lake with the highest kinetic energy (50714 GJ) among all
other source avalanches. This head-end impact, coupled with its high energy, facilitates the
direct forward propagation of waves toward the frontal outlet, causing the lake water to overtop
the frontal moraine and resulting in a higher peak and total discharge (Fig. S7). Thorthormi
Tsho is roughly crescentic in shape and curves toward the west, with its maximum curvature
facing the mass movement origin of loc-6. This shape also allows the impact wave generated
from mass movement from loc-6 to move almost unimpeded along the flow line, resulting in
greater GLOF discharge. In contrast, the direct wave of impact generated by the mass
movement from loc-3, located on the slope above the right moraine dam, is deflected towards
the left lateral moraine, and only a secondary wave proceeds towards the lake outlet, resulting
in a comparatively lower peak and total discharge (Fig. S7). This finding implies that the
geometry of the glacial lake and the surrounding source slope plays a vital role in GLOF output.
Thus, we underscore the importance of considering catchment shape in GLOF modelling,
although we cannot assume that two identical basins will have the same flood properties due
to the influence of other factors, such as the involved volume of solid and fluid parts.
Earlier studies (Mergili et al., 2017; Mergili et al., 2020b) have explained the interaction
between landslides and reservoirs (lakes) and their influence on the resulting hydrograph.
However, these studies did not consider the variables such as directions and angles from
which the mass impacts the lake. To fill this gap, here we enhanced our understanding of the



interplay between the resulting GLOF magnitude and avalanche mass attributes including the
direction and angle from which the avalanche mass enters the lake, the amount of kinetic
energy the avalanche mass possesses and the geometry of the lake. Our results emphasize
the significant impact on the resulting GLOF events caused by the uncertainty in pinpointing
the specific location of origin of mass movement into the lake. Thawing of permafrost and
destabilization of the slope surrounding the lake due to climate warming (Gruber et al., 2017;
Kääb et al., 2018) combined with the expansion of the glacial lake towards the mountain flank
(Rounce et al., 2016) are likely to increase the frequency of mass movement into the lake,
further exacerbating this uncertainty. Therefore, our finding here will be useful to further
improve the development of scenario-based approaches to GLOF modelling (Gaphaz, 2017;
Sattar et al., 2021a) including, high, medium, small and worst-case scenarios (Allen et al.,
2022; Gaphaz, 2017).
**5.3 Mass movement volume, grain density, and entrainment coefficient**
Our investigation revealed that variation in GLOF magnitude is most sensitive to the volume
of avalanches entering the lakes. It also exhibits a significant level of sensitivity to the
entrainment coefficient whilst the grain density ($\rho S$) exhibits negligible impact. For example,
the variation of avalanche volume between $1 \times 10^6$ m$^3$ and $10 \times 10^6$ m$^3$ leads to peak and total
discharge fluctuation of 1160% and 2500%, respectively, and subsequent variation in
maximum flow height and arrival time (Fig. 5 to Fig. 7). The dominant impact of avalanche
volume and entrainment coefficients on GLOF magnitude could be due to their direct influence
on the overall magnitude and intensity of flood events. The total discharge during the GLOF
cascade event is a function of the volume of the avalanche entering the lake. This is further
corroborated by the near-perfect linear relationship between peak discharge ($R^2 = 0.99$) and
total discharge ($R^2 = 1$) with the volume of avalanches entering the lake observed here.
Likewise, the volume of solid content in the flow is solely contributed by the entrainment of
frontal moraine material, primarily determined by the entrainment coefficient ($C_E$). Additionally,
this correlation could be attributed to the amount of energy and associated momentum of the
flow, which changes significantly with corresponding variations in avalanche volume. Also, it
could be due to the longer timing and duration of the flow as evident in Fig. S4. Most GLOF
events in high mountains across the HMA and other alpine regions are caused by moraine
dam breaches triggered by mass movement entering the lake from the surrounding mountain
flank (Shrestha et al., 2023; Lützow et al., 2023; Emmer and Vilímek, 2014). As a result, mass
movement volume is considered a primary basis for scenario development (Allen et al., 2022).
Thus, we believe this finding provides useful insights towards improving the developing of



different scenarios of GLOFs with higher confidence, or is a basis for ensemble testing, with
the caveat that the range of outputs may be too wide to be of practical use.

**5.4 Frictional parameters variations**

Among the frictional parameters, our result showed that GLOF magnitude is most sensitive to
the δ. For example, the variation of total discharge (47.5%) resulting from fluctuation of δ within
the conservative range was 30 times greater than that of internal friction angle ($\phi$) and over
four times greater than that of fluid friction angle ($C_{FF}$). δ plays a dominant role in flow dynamics
and the interaction between the flowing material and the channel bed. This direct contact
means that even minor changes in δ can have substantial effects on the resistance
encountered by the flowing material, thereby influencing the mobility of the flow (Pudasaini
and Krautblatter, 2014b; Mergili et al., 2018a; Mergili et al., 2018b). $\phi$ on the other hand
primarily affects particle interactions within the flowing material, whilst $C_{FF}$ is a coefficient which
quantifies the overall flow resistance within the flow path mainly depending on surface
roughness. Our findings indicate that prioritizing the consideration of δ over the other two
frictional parameters is advisable. This can be done by determining spatially variable values
through field data or conducting a statistically substantial sensitivity analysis. Nonetheless,
despite the relatively low overall impact on GLOF magnitude, the CFF notably increased the
flow's mobility, especially beyond 12 km downstream, when the flow became fluid-dominant
(Fig. S4).  This because CFF is controls the mobility of the fluid part (Mergili and Pudasaini,
2024; Mergili et al., 2017). This suggests that $C_{FF}$ could exert a substantial influence,
particularly in modelling scenarios encompassing longer flow distances.

**5.5 Key points from the comparison of all parameters and the way forward**

Identifying the most accurate parameter values or optimal datasets can be achieved through
validation with well-constrained historical events (Zheng et al., 2021a; Schneider et al., 2014;
Mergili et al., 2020b; Shugar et al., 2021), but there are limitations in the transferability of these
findings due to the unique characteristics and initial conditions of each GLOF, such as varying
volumes of solid and liquid. These specific conditions mean that the results of one modelled
GLOF event might not accurately predict the behaviour of GLOFs in different regions or under
different circumstances (Mergili et al., 2018a; Mergili et al., 2020b). Therefore, while these
back-analysed parameter values can provide valuable insights, they need to be applied with
caution and adapted to the specific context of each new GLOF scenario. This is emphasized
by our finding that the characteristics of the modelled GLOF are substantially impacted by
various parameters. As a result of these multiple sources of uncertainty in modelled GLOF, it
could pose challenges in effectively communicating risks with communities and other



stakeholders (Thompson et al., 2020). We highlight that more sensitive parameters should be
treated carefully in future GLOF modelling works by robustly considering associated
uncertainties.
Due to the high sensitivity of the model output on DEM resolution, we emphasize the critical
importance of high-resolution and good-quality DEM (Uuemaa et al., 2020; Schumann and
Bates, 2018), especially when modelling is aimed at producing hazard maps with higher
granularity at the specific basin scale.  Specifically, DEMs should be the high spatial resolution,
high vertical accuracy and recently produced, especially in areas of high relief and rapid
landscape change such as in Himalaya (Schumann and Bates, 2018).  Previous studies have
indicated that flood modelling accuracy can be improved by correcting the effect of DEM
resolution and accuracy (Saksena and Merwade, 2015) or by merging with other high-
resolution and accurate DEMs (Muthusamy et al., 2021). These methods appear viable in the
context of highly sparse coverage of high-resolution DEMs and the unaffordability of high-
resolution commercial DEMs, but the modelling results should still be interpreted with caution.
On the other hand, whilst it poses computational challenges, especially with high-resolution
DEMs, we believe that selecting a mesh size equivalent to the spatial resolution of the DEM
could effectively mitigate uncertainty associated with mesh size variation. Models such as D-
Claw which features patch-based adaptive mesh refinement capability can be potentially used
as alternative models, however, its use in GLOF modelling is limited so far (Iverson and
George, 2014; George et al., 2017).
Avalanche volume and $\delta$ exhibit a strong linear relationship with all output parameters. Whilst
the linear relationship does not negate the influence these parameters have on flow
characteristics, it suggests that model output errors resulting from uncertainties in these
parameters might be predictably managed. This is essential since predicting the volume of
mass movement involved in the forward modelling is highly challenging and determining an
accurate value is impossible – the current challenge is rather to establish a likely envelope of
volumes. However, such prediction should be bespoke to the particular events based on the
initial parameters like estimated ice thickness, slope, and presence of permafrost.
Furthermore, such predictions must also consider other factors, such as equifinality arising
from the interaction of multiple parameters (Mergili et al., 2018a; Mergili et al., 2018b; Mergili
et al., 2020b).
The $C_E$ exhibits a linear relationship only with volume. This relationship with the volume is
understandable, as the entrainment coefficient is a primary determinant of how much solid
fraction of the flow is added due to erosions. However, the arrival time exhibits distinct
thresholds at the entrainment coefficient $10^{-6.46}$ kg$^{-1}$.  The decrease in flow arrival time



observed until a CE value of $10^{-6.46}$ kg$^{-1}$ may be attributed to the flow being primarily dominated
by the fluid component, with the contribution from erosion being negligible. However, the
subsequent increase in flow arrival time as the CE value further increased from $10^{-6.46}$ kg$^{-1}$ to
$10^{-5.85}$ kg$^{-1}$ could be attributed to the effect of increasing concentration resulting from a higher
rate of erosion. This suggests that once this threshold is surpassed, the resulting peak flow
and arrival time demonstrate a heightened sensitivity to variations in entrainment.
Consequently, this sensitivity may translate to the flow characteristics such as flow height and
arrival and arrival time which are essential for hazard and risk assessments. It is important to
note that this threshold value may vary across different GLOF events due to the diverse
combinations of other parameters.
The linearity demonstrated by the initial volume of avalanches entering the lake and $\delta$ warrants
further investigation into flow characteristics resulting from variations in these parameters.
Further investigation with adequate sample sizes and a reliable statistical approach would
enable the establishment of accurate relationships or predictor values. The threshold effect
observed in the $C_E$ value also warrants further investigation using statistically conclusive
samples to determine whether the threshold value is universal across different events or
specific to individual occurrences. For factors such as internal friction angle, fluid friction
number, and $\rho S$, the conservative values may suffice or receive less emphasis, particularly
considering the numerous parameters involved in GLOF modelling.
The r.avaflow model provides comprehensive and open-source codes for simulating
cascading mass flow in complex topographies (Mergili and Pudasaini, 2024). Its
comprehensiveness stems from the wide range of parameters it considers, making it a
versatile tool for various mass flow process chain simulations (Mergili et al., 2017). Past
studies have demonstrated the model's ability to accurately back-calculate historical events
with detail (Shugar et al., 2021). However, challenges persist in its application to forward
modelling (Allen et al., 2022; Sattar et al., 2023), particularly in the context of GLOF hazard
and risk assessment (Mergili et al., 2020b). In our study, we conducted a robust sensitivity
analysis considering nine parameters relevant to GLOF towards addressing these challenges.
Since we identified the key parameters that significantly influence the modelled GLOF output,
our result can be used as a basis for further improvement and optimization of r.avaflow
modelling codes.
The GLOF simulations were conducted using the r.avaflow model due to its capability to model
the entire GLOF process chain (Mergili and Pudasaini, 2024; Mergili et al., 2017). While we
present the uncertainty involved in the full process chain GLOF from mass movement entering
the lake to downstream propagation, we specifically explored the uncertainty of the GLOF



input parameters relevant to r.avaflow modelling. Input parameters such as DEM datasets,
and the volume and density of mass movement involved in a GLOF event, might be similar
across different models. However, we caution that the parameters tested here do not
necessarily apply to all models used for GLOF modelling.
The flow arrival time was measured from the profile located 3 km, 6 and 9 km downstream of
the lake since some of our modelled GLOF terminates before proceeding further downstream.
This is a reasonable point as human settlement downstream of the lake is mostly concentrated
around this area. The variation of flow arrival time might be underestimated if the location is
farther downstream from the lake.
Here we focused on nine essential parameters in r.avaflow, which are relevant to GLOF
modelling. However, including inbuilt modules, initial conditions, and all flow parameters,
r.avaflow has more than 30 parameters (Mergili and Pudasaini, 2024) (Table S1). Thus, our
sensitivity analysis might have potentially overlooked the complexity of r.avaflow stemming
from the effect of all these parameters.
One-at-a-time sensitivity analysis we used here, inherently lacks consideration for parameter
interactions and may have potentially overlooked important relationships (Saltelli et al., 2004).
Moreover, due to the immense computational cost of r.avaflow, we used only 10 simulations.
While this number of simulations for each parameter produced substantially conclusive
results, we do not discount the robustness of global sensitivity analysis employing an adequate
sampling size. Future studies should focus on testing further r.avaflow parameters and in-
depth model analysis by employing a statistically sufficient sampling size.
**6 Conclusions**
GLOFs present substantial dangers to communities residing in valleys downstream of glacial
lakes. GLOFs involve complex cascading processes and typically occur across rugged
mountain terrains. Due to these complexities, modelling GLOFs necessitates extensive input
data, parameters, and complex modelling codes for accurate hazard and risk assessments,
which is inherently challenging. However, previous studies have mostly relied on open-access
data and are grounded in a historical event introducing significant uncertainties to the
modelling results. In this study, we have, for the first time, conducted sensitivity analysis
considering multiple GLOF parameters and ranked these inputs based on how their
uncertainties in input values apportion to the variation in modelling output, by employing
cutting-edge modelling code, r.avaflow.  Our results suggested GLOF modelling outputs such
as peak and total discharge are substantially sensitive to variation in input values of six out of



nine parameters we tested here. Specifically, the modelling outputs are the most sensitive to
the volume of avalanches entering lakes followed by the variation in DEM datasets and the
location of origin of mass movement entering the lake. Other parameters like mesh size, basal
frictional angle, and entrainment coefficient also showed significant sensitivity. Although
limited to GLOF modelling with the r.avaflow model, our study emphasizes that GLOF
modelling results are influenced by uncertainties stemming from various sources,
underscoring the need for careful interpretation of the modelling results. By ranking the model
parameters according to their impact on model output, our study prioritizes model input
parameters for future modelling efforts, given the challenge of adequately constraining multiple
parameters. Additionally, this study lays the groundwork for a thorough investigation into the
most sensitive parameters, to improve our understanding of GLOF modelling.
**Acknowledgement**
This work was supported by the Natural Environment Research Council (NERC)- funded
IAPETUS Doctoral Training Partnership [IAP2-21-267]. We thank Dr. Sonam Wangchuk for
helpin us in setting experiment in the HPC.
**Code and data availability**
The r.avaflow modelling code we used here for simulating all scenarios of GLOF can be
accessed at: r.avaflow | The mass flow simulation tool (landslidemodels.org). The SRTM GL3,
NASADEM and AW3D30 DEMS used here can be downloaded from the OpenTopogragphy
at: OpenTopography - Find Topography Data. The HMA-DEM can be downloaded from the
National Ice and Snow Data Center at: High Mountain Asia 8-meter DEM Mosaics Derived
from Optical Imagery, Version 1 | National Snow and Ice Data Center (nsidc.org).
**Supplement**
The supplement related to this article is available online at:
**Author contributions**
SR, SA. and RC conceptualized the study. SR undertook the computational studies and data
analysis. AS provided guidance in modelling. MM revised and provide expert opinion on the
study. SA, RC and AS supervised the work. All authors wrote and edited the manuscript.
**Competing interests**
The contact author has declared that none of the authors has any competing interests.



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
