# Peer review of "Exploring implications of input parameter uncertainties on GLOF modelling"

_EGUsphere, 2024_

## Author Comment (AC1)

Reviewer one

General Comments:

This study conducted a sensitivity analysis of major parameters of the GLOF numerical simulation using the open-source software r.avaflow model to determine which parameters significantly affect outputs closely related to GLOF hazard and risk assessment, such as peak discharge, total discharge, flow arrival time, and reach distance. The manuscript is well-organized, and the references are appropriate, demonstrating the research's significance and originality. All figures are of good quality, and the information is clearly presented.

I have commented on some points requiring minor revision. Please refer to the specific comments below for detailed suggestions for revision.

Many thanks for the positive review of our manuscript and valuable feedback. we sincerely appreciate the time you have taken to review it. Below, we provide a point-by-point response to the reviewer's comments, with our response highlighted in blue font for clarity.

Specific Comments:

**1    l.86-89: Since the previous sentence already states that r.avaflow is open source, it would be better to delete "is open source and" from this sentence and instead write, "r.avaflow allows modification of all input parameters, ...".**

Corrected as suggested.

**2    l.159: The text refers to "Figure 1B and 1E," but the subfigure labels are written in lowercase. Please make the labelling consistent throughout the manuscript. The same correction applies to the following text, figures, and supplement.**

Corrected as suggested. All the signposting to the figures now is done in lowercase consistent with the figure labelling in the revised manuscript.

**3    l.165: If you use scientific notation, it might be more appropriate to write $300 \times 106$ m3 as $3 \times 108$ m3.**

Amended as suggested.

**4    l.171: Insert "shows" or another appropriate verb after "The map (a)."**

Amended as suggested.

**5   l.254, Figure 2: There is no legend for CFF in the figure.**

The legend for the CFF is added as suggested.

**6   l.287: It would be better to write this as 106-107 m3.**

Amended as suggested for this one and in all appearances.

**7   l.310: It says, "3.2.1 Digital elevation (1) and mesh size (2)," but some of the subsection titles from 3.2.1 to 3.2.6 include numbering, while others do not, which causes confusion. It might be better to remove the numbering from the subsection titles altogether.**

Initially, the numbering was done to make it convenient for the reader to refer to section 3.2, where parameters are introduced. In the revised manuscript, we have removed all the numberings after the subheadings as suggested by the reviewer.

**8   l.325: The unit should be written as kg m-3. Similar unit errors are seen from this point onwards.**

The unit is corrected as suggested, as are all subsequent appearances.

**9   l.329: It would be better to write this as 2.94×108 m3.**

Corrected as suggested.

**10   l.428-430: Table 3 does not show the results described here.**

The signposting to Table 3 is deleted. Thanks for pointing out this typo. We have added the supplementary figure (Fig S3) for the arrival time along the river profile line and signposting here and subsequent appearances are amended accordingly.

**11   l.431: It would be better to insert the unit (min) after the number 4.37.**

The unit is added as recommended.

**12   l.434-437, Figure 3: The figure title indicates that the left and right panels respectively show maximum flow heights and hydrographs, but this is the opposite. Additionally, the left panel of each subfigure (a)-(d) does not have a horizontal axis label. Also, it is not clear what the dashed lines in the right panel indicate.**

Thanks for the careful spot. The figure panel labelling is amended as suggested. We have also added the x-axis label for the left panels. The solid line represents the flow height of the liquid, and the dashed line represents the solid part of the flow. These captions are now updated in the revised manuscript.

**13    l.440-441: Although this description refers to Fig. 3, it is difficult to interpret the flow velocity characteristics from Fig. 3.**

After revisiting the figures and analysis we have now amended lines 440-441 to 'resulting in a significant increase in flow characteristics like flow depth' which is consistent with data indicated by Figure 3.

**14    l.456, Table 3: What does the "SL no." in the leftmost column stand for?**

We meant to say, 'serial number.' But we have amended it from 'SL no.' to 'No.' to avoid any confusion in the revised manuscript.

**15    l.470-471: Is it not possible to determine from Fig. 4 how the solid volumetric portion exhibit fluctuates?**

Thank you for pointing out this. We have now signposted to supplementary figure 3, where we added the flow height of debris and fluid resulting from the GLOF initiated by a mass movement entering from different directions.

**16    l.486, Figure 5: What does the light-colored shaded area in each subfigure represent (e.g., 95% confidence interval)? It would be better to clarify this in the legend or caption. Additionally, the regression equation and the coefficient of determination seem to mismatch in some subfigures. Please verify to ensure there are no errors. These comments also apply to Figs. 6 and 7.**

Thanks for pointing this out. Yes, the light colour-shaded region shows a 95% confidence interval. We have now mentioned this in the caption. Coefficient and regression equations are corrected in the revised manuscript.

**17    l.493, Figure 6: The label on the vertical axis should be replaced with "total discharge."**

The y-axis label is amended to 'Total discharge' as recommended.

**18    l.533: From this point onwards, the CV value is written in units of %, but isn't it common to treat CV as a dimensionless value if it is derived by dividing the standard deviation by the mean? If the coefficient of variation is expressed as a percentage, it would be better to clearly state this in the text.**

Thanks for pointing out this. We agree that CV is a dimensionless value. Here we multiplied the CV value by 100 to present the CV in percentage form. However, as recommended, we have now clarified this in line 410 as 'We multiplied each CV value by 100 to express it in percentage form'. And amended line 530 from 'calculated the coefficient of variation to 'percentage coefficient of variation'.

**19    l.538-: The following descriptions of the regression analysis results show some differences between the coefficients of determination shown in Figs. 5-7. Please check and correct as necessary.**

Thank you for pointing out this. We have noticed a typo in the figures in the earlier version. The values for $R^2$ are corrected in the revised manuscript.

**20    l.548: Insert "(Fig. 6)" after "total discharge."**

Amended as suggested.

**21    l.558: It states that the volume range is 4×106 m3 to 10×106 m3, but the figure shows normalized parameter values, making it difficult to correlate with the actual volumes. It would be better to include the normalized values here. The same comment applies to the description of CE in l.560-562.**

Thanks for pointing out this. To avoid confusion here and in all appearances, we have changed the x-axis scale of Figures 5 and 6 to the original value of the parameter we used in modelling in the revised manuscript.

**22    l.585-586: The description "ranging from HMA-DEM (8 m) to SRTM GL3 (90 m)" would be better described as "with resolutions ranging from 8 m (HMA-DEM) to 90 m (SRTM GL3)."**

Amended as suggested.

**23    l.686: This is the first time mention of δ in the discussion section, so it would be better to use the notation "basal friction (δ)" as with the other parameters.**

Amended as suggested.

**24    l.700: This is likely a mistake; it should read, "This is because CFF controls the mobility of the fluid part." Also, it would be better to write the "FF" in CFF as a subscript throughout the text.**

Thanks for pointing out the typo. The sentence is corrected as suggested. We however chose to keep the short form of fluid friction as CFF as it is in the r.avaflow manual.

**25    l.833: "helpin" should be replaced with "helping."**

Amended as recommended.

---

## Author Comment (AC2)

Dear Editors,

Dear Authors,

Thank you for giving me the opportunity to review manuscript egusphere-2024-1819, "Exploring implications of input parameter uncertainties on GLOF modelling results using the state-of-the-art modelling code, r.avaflow" by Sonam Rizin and co-authors. In this study, the authors investigate the sensitivity of key modelling outputs (peak discharge, flood volume, and flood arrival time) in r.avaflow to variations in nine selected input parameters. r.avaflow is a widely used software for simulating catastrophic mass flows, making this analysis highly relevant for researchers and practitioners interested in modelling glacier lake outburst floods (GLOFs). The authors conclude that the volume of landslide material entering the lake has the greatest influence on simulation outcomes, followed by the digital elevation model (DEM) and its resolution, while other parameters show comparatively lower impacts.

This study offers valuable insights for users of r.avaflow, especially those interested in identifying the parameters that exert the strongest control over GLOF modelling results. However, several aspects of the study require clarification and further discussion to strengthen its contributions.

Many thanks to the reviewer for the careful evaluation of our manuscript and for providing us with very detailed feedback. Below, we provide a point-by-point response to the reviewer's comments, with our response highlighted in blue font for clarity.

Major Comments

1) Parameter Selection and Exclusions

While focusing on a subset of parameters is practical given the complexity of r.avaflow, the rationale for selecting exactly these nine parameters remains unclear. For instance, the authors treat lake bathymetry and volume as constants, yet these are highly uncertain and challenging to estimate, particularly for remote glacier lakes. Would a shallower lake generate a higher displacement wave, potentially resulting in a larger peak discharge? What about the height of the moraine dam and a potential bedrock sill beneath it? How would this (not uncommon) setting change entrainment and accordingly, peak discharge, once that bedrock sill is hit? Similarly, the study does not examine the effects of varying the velocity or grain size of the landslide entering the lake. These factors may influence wave dynamics and warrant at least a discussion in the context of the available literature.

Moreover, with r.avaflow offering more than 30 tunable parameters, it would be helpful to understand in more detail whether the excluded parameters were found negligible or simply beyond the scope of this study. While a comprehensive sensitivity analysis of all parameters may be impractical, a broader discussion of the omitted parameters' potential roles would add value to the manuscript.

We sincerely thank the reviewer for this thoughtful feedback. The r.avaflow modelling code has 30+ tuneable parameters and is open-source, which sets it apart from many modelling codes, where most of the parameters remain hidden within a 'black box'. However, as rightly noted by the reviewer, conducting sensitivity analysis of all these input parameters is impractical and beyond the scope of this study due to the extensive computational time and resources required for the model. We selected these nine essential parameters, which are identified as important in several previous studies and frequently manipulated in the context of GLOF modelling to fit with back-calculated parameters (Vilca et al., 2021, Zheng et al., 2021), making it critical to evaluate their impacts on model outputs. While some previous studies have provided sensitivity analysis for a subset of these parameters - basal friction angle and entrainment coefficient (Mergili et al., 2020, Baggio et al., 2021), we have enhanced the robustness of sensitivity analysis by expanding the parameter set to nine and conducting multiple simulations. However, acknowledging the concerns raised by reviewers, in the revised manuscript, we have added rationale for selecting these parameters in the introduction (lines 149 to 153) and added this concern in the discussion (lines 834 to 847).

We appreciate the reviewer for highlighting examples of important input parameters. Recognizing the reviewer's concern and the significant influence of lake volume on the modelled GLOF output, we will incorporate this parameter into the revised manuscript.

Regarding the height of the moraine and the presence of a bedrock sill, we acknowledge their significant influence on modelled GLOF output. However, we opted not to include these parameters for two main reasons. First, variations in the entrainment coefficient serve as a surrogate for these factors, as changes in this coefficient can capture variations in moraine erosion and the resulting GLOF output. Second, we believe that the height of the moraine can be effectively constrained using high-resolution DEMs and/or fieldwork, in contrast to other parameters, e.g. the source and volume of avalanching material entering the lake, which are far more challenging to quantify. Additionally, we did not include avalanche grain size in our analysis, as r.avaflow does not currently support this capability. However, concerning avalanche velocity, we understand its importance, but the GLOF process chain we modelled here are scenarios involving a rapid mass movement process for which the frontal velocity and frontal height are inherently interdependent. So we believe the velocity of an avalanche

entering the lake is inherently determined by the characteristics of avalanches we assumed such as grain size, volume, basal and internal friction and terrain conditions (which we believe is being taken care of by considering varying avalanche locations).

2) Parameter Value Ranges and Physical Plausibility

The ranges of parameter values used in the sensitivity analysis appear to be informed by prior studies, but it is unclear how well these values indeed reflect the physics of GLOFs. For instance, are the chosen ranges realistic for a variety of glacier lake settings? While the authors acknowledge the challenge of equifinality—achieving the "right" results for the "wrong" reasons—this issue is amplified by the absence of validation against real-world cases.

Applying the sensitivity analysis to a documented GLOF event, such as those at Langmale, Salkantay, Elliot Creek, Ranzeria Co, Chongbaxia Co, or Tam Pokhari, could provide a much-needed validation framework. This would allow the authors to test whether the chosen parameter ranges lead to realistic flood scenarios and to assess how uncertainties in key parameters (e.g., landslide volume) translate into variability in flood predictions.

Many thanks to the reviewer for pointing out this. The reviewer's concern about the parameter ranges and their applicability to the glacier lake in different environment settings including their application to well-known events is the fundamental motivation for our study. Specifically, we want to understand how variation in input parameter values, within commonly used ranges, influences the GLOF modelling results, instead of trying to determine the 'correct' value for each parameter. Or, put another way, if we apply our study to a specific example, we may determine that certain factors are more important than others for this specific example, but it would be unclear how applicable our results are to other events. Thus, we see our approach as the least biased towards any particular event and hence the most generally applicable approach. We have amended our discussion lines from 833 to 842 to make this point very clear and motivation is mentioned in the introduction (lines 142 to 148) in our revised manuscript. Partly, we are motivated to choose Thorthormi instead of some lakes with previous GLOF record as it has been identified as the most dangerous glacial lake in Bhutan (NCHM, 2019). We believe Investigating the GLOF input parameters specific to this lake could provide valuable insights for future studies aimed at better understanding the hazard and risk it poses besides merely understanding impact input parameters on GLOF modelling in general.

3) Interactions Between Parameters

The current analysis isolates each parameter's effect by varying one parameter at a time. While this approach is useful for identifying individual sensitivities, it does not account for potential interactions among parameters. For example, do certain combinations of parameter changes amplify or mitigate the overall effects on model outputs? Exploring such interactions is crucial for providing a comprehensive understanding of the system dynamics and would greatly enhance the applicability of the study's findings.

Many thanks to the reviewer for highlighting this important consideration. We were aware of the influence of parameter interactions in GLOF modelling results, however, the extensive computational resource and time required for running each model means it is not practical to assess parameter interactions within the scope of this study or the broader research project. To illustrate this point, one model run takes around 10 to 30 hours, depending on the initial conditions, so running 10 different values for each of our 9 parameters takes 900 to 2,700 hours. If we then do each combination of interactions, it will require a very large amount of computing time and power. However, while we cannot afford to conduct model parameter interaction here, acknowledging the reviewer's concerns, we have now discussed the importance of considering the model input parameters and how they can be addressed in future including the feature that could enable parallel computing (lines 811 to 836).

4) Practical Recommendations for Model Users

The manuscript would benefit from more confident and actionable recommendations for users of r.avaflow. Currently, the authors caution against overconfidence in interpreting parameter values but stop short of providing concrete guidance. A "starting point" for parameter selection or a framework for iterative refinement would be invaluable for new users. Additionally, the discussion could include recommendations for scenarios where multiple parameters are varied simultaneously, which more closely reflects real-world uncertainty.

The reviewer's concern regarding the need for practical recommendations for r.avaflow users echoes our goal of this study, which is to provide insights into the uncertainties of GLOF modelling results. Unfortunately, it is tricky to provide any concrete and prescriptive recommendations for future users. However, we have discussed how each of the parameters in the future study in lines 747 to 797.   We will make this discussion very exclusive in our revised manuscript. Similarly, providing practical recommendations for scenarios involving simultaneous variations of multiple parameters is not feasible at least for this study as we have emphasized the significant role of parameter interactions in contributing to modelling uncertainties and have provided clear recommendations for future studies to focus on these interactions in the discussion as mentioned in point 3.

5) Clarity and Presentation

Finally, I suggest a careful revision of the text and figures to improve clarity and polish. For example, the abstract should be revised to better summarize the study's scope and relevance, the study area, and implications. Specific suggestions for improvement are provided below.

We have revised the next version submission after careful proofing and amending all figures as suggested.

Specific comments

L2: The abstract should explain why r.avaflow represents the "state-of-the-art" in GLOF modelling, otherwise please consider removing this phrase.

"state-of-the-art" is removed in the revised manuscript in all appearances.

L9-13: Please try to express one idea per sentence. This opening sentence has at least three, while also including some confusion. As far as I understood, this study does not include direct measurements, which seem to be a core motivation in both forward modelling and back analysis?

Thanks for the suggestions. The sentence is corrected as suggested.

L15: How many different GLOFs did you assess in these 78 simulations? What is the key criterion that you evaluated the suitability of the model? Some kind of intersection over union between mapped and simulation runout areas or flow depth?

We appreciate the reviewer for pointing out this. We would like to clarify that each simulation represents one GLOF scenario. This is being clarified in the revised manuscript as '88 simulations each representing a unique GLOF scenario'. We have changed 78 to 88 since we are going to include volume variation in the revised manuscripts. We would like to ensure that each model set-up was carefully evaluated for stability.

L16: Please add a motivation why r.avaflow was selected among the many available mass flow models.

We selected the r.avaflow model firstly because it can model the full process chain involved in landslide-triggered GLOF, which is the most common form of GLOF in High Mountain Asia. Second, it is open source. Most importantly, r.avaflow modelling code allows users to

manipulate all parameters which is key for this study. This transparency sets it apart from many modelling codes, where most of the parameters remain hidden within a 'black box'. We acknowledge that this has been not made clear. While we are not able to add this in the abstract due to the word limit, we have added this in introduction lines 86-89.

L17: You need to introduce which GLOF exactly you model, as certainly not every GLOF is triggered by a mass movement entering the lake. It's also important to emphasize the dam type: moraine-dammed, bedrock-dammed, or a combination of both?

Thanks, reviewer for pointing this out. We have corrected it as "mass movement-triggered moraine-dammed GLOF modelling" in the revised manuscript.

L17-18: I was surprised not to see lake depth/ volume/ bathymetry in this assessment. Doesn't the – in many cases unknown – depth of the lake play a fundamental role on the amount of water that can be pushed out of the lake?

We agree that the volume of the lake is one of the most important parameters in GLOF modelling. We have added volume to the revised manuscript. The number of parameters will be accordingly changed to 10 and change in conclusion will be mention in the revised manuscript.

L16 and 20: What is 'GLOF output'? Please rephrase and explain.

Corrected as "GLOF output parameter".

L19-21: Somehow repetition of the preceding sentence. You could just add the CV for every variable in the preceding sentence, which could create a bit more space for other findings of your study.

Thanks. Corrected as suggested. The corrected sentence now reads: "The GLOF output parameter resulting from the volume of mass movement impacting lakes has the greatest coefficient of variation (CV) = 47%, while the internal friction angle had the lowest CV (0.4%)."

L24: Unclear what you mean with 'statistically'?

Amended to "We recommend that future GLOF modelling should carefully consider the output uncertainty stemming from the sensitive input parameters identified here, some of which cannot be constrained before a GLOF and therefore must be addressed using statistical approaches."

L26-30: Consider updating these numbers with the global glacier lake inventory presented by Zhang et al. (2024), Communications Earth and Environment, as those presented by Shugar et al. 2020 are subject to large errors.

Many thanks to the reviewer for pointing out this. We are aware that Zhang's data is latest, but we are trying to show here the volume of water stored in the lake, which is important from the modelling perspective.

L28: Seems that this study focuses on glacial lakes in HMA, so please state this here.

Thanks for reminding this. We have now mentioned HMA.

L32: replace 'mass inputs' with 'steep slopes'?

Corrected as suggested.

L33-36: Statements about GLOF frequency seem a bit out of place here, as you are describing the physical process of GLOF triggering before and after?

Removed as suggested.

L46: Remove 'the' in front of HMA.

Corrected as suggested.

L50: Consider avoiding subjective terms such as 'unfortunately'.

Removed "unfortunately" as suggested.

L58: Consider adding a note that dams not necessarily fail completely?

Thanks, amended as follows: "However, it is important to note that in some cases, these triggering factors may not necessarily result in a complete moraine dam failure".

L63: Including the availability and entrainment of sediment and its grain size distribution?

Thanks for the suggestions: the statement is now corrected as:

"As the flood propagates further downstream, it can transform into a debris flow and/ or a hyper-concentrated flow depending on the geologic and topographic characteristics of the

river channel as well as depending on the availability of erodible sediment and its grain size distribution".

L72-73: Either 'most' or 'all'

Changed to 'most of'.

L75: 'sediment' entrainment?

Amended as suggested.

L79: Consider adding 'into Imja Lake, Nepal'

Amended as suggested.

L96: What 'model outputs' does r.avaflow provide?

We have now modified the statement to:

"However, the precision of these model output parameters such as peak flow, depth and velocity depends on the accuracy of various input parameters and initial conditions, including the release height of mass, the resolution and vertical accuracy of the digital elevation model (DEM), density, entrainment, and frictional parameters".

L105: Does the velocity of the mass movement entering the lake also play a role?

We agree with the reviewer's concern about the velocity. This is now addressed as noted above.

L116: 'data acquisition'?

 Amended  to 'acquisition'.

L119: What kind of 'data'? Gridded ice thicknesses from ice-flow inversion models?

Thanks. Corrected as "ice thickness data from the ice flow inversion models".

L120: I would rather deem these 'arbitrary' ice thicknesses.

Thanks, the word 'conservative' is changed to 'arbitrary' as suggested. However, we did not add 'ice thickness' here because it refers to all types of mass that might hit the lake, including landslides, rockfall and ice.

L124: My background is more in statistics, where parameters (coefficients) are usually distinguished from variables (predictors). It would be good if you could define what a 'parameter' is in your study because sometimes I have the impression that you are talking about model coefficients or constants that are free to change or optimize in r.avaflow, rather than the input datasets, which I would rather call a variable.

Thanks for your clarification about the model parameters. We now defined input parameters (in lines 282 to 286) as:

"In the context of r.avaflow, a parameter is an (often user-defined) variable influencing the physical characteristics of the movement or the numerical behaviour" of the flow. Parameters can be based, e.g., on physics (such as friction angles) or empirical knowledge. Parameters can be represented by global values, by individual values for each raster cell, or by time-dependent values."

L135: Does 'employing' have the same meaning as 'inferring' here?

Thanks for this suggestion. We have now changed to 'inferring'.

Table S1 is labelled Table S4 in the Supplementary Material. Please correct. In any case, I found the choice of these nine parameters a bit brief and poorly referenced. It seems that you keep all the 29 other parameters in your experiments fixed. So, one might argue that playing around with these 29 parameters could equally reproduce our outcomes if keeping your selected 9 parameters fixed. Please add more motivation why these 9 are so much more important than the others.

Thanks for this valuable feedback. The supplementary table caption is amended. Our selection of nine parameters was motivated by our findings that these input parameters are considered essential and have been frequently adjusted in previous studies to align with values inferred from observed past events. We believe that these parameters are the most likely to influence the results of future modelling efforts. This justification has been added in the introduction (lines 148 to 151).

Figure S1: Please explain the symbols in the caption or in the x-axis. In addition, I wondered if and why it makes sense to stay within the limits of the parameter values reported previous

studies? Do they represent the physically plausible range? Does the local setting match with that represented in your study?

Thanks for pointing out this. We have now explained the x-axis of Figure S1 in the caption. In terms of the parameter value range, we would like to reiterate that the parameter ranges used in our modelling are grounded within the physically plausible range established in various previous studies. Our study aims to quantify the uncertainty that arises by using values within the parameter ranges inferred from a variety of observed events, rather than all possible scenarios. As noted above, there is a significant computational cost to running these scenarios, so we want to ensure that the scenarios we run are most useful to the community and we therefore exclude less likely end members by staying within the published parameter range.

L167: Not sure where you showed the 'high outburst susceptibility'?

Thank you for pointing out this. We have changed the statement from "This high GLOF susceptibility and potential " to "This high GLOF hazard" as the proceeding statements address the GLOF hazard.

Figure 1b: Loc-4 occurs twice? I cannot read the right black label of the yellow dot, starting with "Lo…". What does 'Loc' actually mean? Please add in the figure caption.

Thank you. An abbreviation for Loc-1 to Loc-6 is added in the caption.

L179: add access date of OSM data.

The date is added.

L185 is this infrastructure within the first 10 km downstream of the lake? Or more?

We have added 10 km to emphasize the risk of GLOF for the people located nearer to the lake. However, we acknowledge the confusion it has created so we removed the statement "202 buildings are located within the immediate 10 km downstream of Thorthormi Tsho " in our revised manuscript.

L189: Check grammar.

Thanks for the suggestion. The sentence is corrected as: "Also, the Punakha Dzong, which has great historical and cultural significance to Bhutan is located downstream of Thorthormi Tsho."

L225: What does 'are scaled with a solid fraction of the flow material' mean? How do you know the flow material? Please explain.

Thank you, reviewer, for this feedback. Here we meant to say that both the internal and basal friction angles are adjusted based on the solid fraction of the flow material. This adjustment ensures that the model accurately reflects how the flow's frictional properties change depending on its composition. This feature in r.avaflow s  is disabled by default but needs to be enabled if this dynamic evolution of frictional parameter is to be capture.  However, we did not use these functions in r.avaflow in our modelling work. So, we have now deleted these sentences from the text to avoid confusion.

Figure 2: why do you show here a distinction between H, M, and L scenarios? I guess this stands for High, Medium, and Low, but this appears neither in the figure caption, no in the manuscript, no?

Thanks for pointing out this. We have removed these from the figures in the revised manuscript.

L258: This reference (Mergili and Pudasaini, 2024) is not part of the list of the list of references, but what I understand is that this is a link to a website? https://www.landslidemodels.org/r.avaflow/direct.php

Thanks. The Mergil and Pudasaini, 2024 is added to the reference list.

L287: Are these landslide volumes representative for previous landslide-generated lake outburst floods? I still wonder how much the landslide velocity modulates the displacement of water. If the landslide enters slowly (<1 m s-1) the water body, would you expect a much lower wave height?

Thank you for highlighting this. We want to clarify that determining volume of mass movement based on earlier studies is difficult due to sparsely known volume of GLOF triggering mass movement.  However, we believe that the volume range of 1 to 10 × 10$^6$ m$^3$ chosen for this study is reasonable. This is because historical mass movement entering lake that caused GLOF are mostly higher than 1 × 10$^6$ m$^3$   (Zheng et al., 2021, Byers et al., 2018) although recent event in Sikkim Himalaya was known to be up to 16.75 × 10$^6$ m$^3$ (Zhang et al., 2024).

Additionally earlier studies in Nepal Himalaya which also calculated avalanche volume based on topographic potential and assumed avalanche thickness has estimated the volume between $2.7 \times 10^4$ to $6.7 \times 10^6$ m$^3$ (Lala et al., 2018, Rounce et al., 2016). Having said this, we acknowledge the importance of robustly considering choice of mass movement volume. Therefore, we have amended section 3.2.2 (lines 351 to 361) in our revised manuscript.

We agree landslide velocities and volume modulate the displacement wave. Two landslides with the same volume originating from the same slope but different velocities (slope effect) would create different impulse wave heights and resulting GLOF as we have addressed in point 1.

L292: Why is that range 'conservative'? What I read from these references is that these values were rather 'informed by the expertise of the model developers'? In my opinion, 'conservative' implies that the selected values rather seek to model the lower bound of potential GLOF discharges to avoid gross overestimates. By contrast, you seem to rather aim for the mean estimate for those parameters by excluding values outside of the interquartile range.

We have removed 'conservative' from the statement and amended to:

 "For parameters 6-9, we gathered various values employed in previous studies (Allen et al., 2022; Mergili et al., 2020a; Mergili et al., 2020b; Vilca et al., 2021) and computed descriptive statistics and established the median, upper quantile value, and lower quantile for each parameter using these collated values".

L314-316: I wondered why you decided not to use the ALOS PALSAR DEM (12.5 m resolution) as a compromise between the high-resolution HMA and low resolution SRTM DEMs?

ALOS PALSAR DEM is a resampled version of SRTM-30 m, so we selected NASADEM as it is the modernised version of SRTM DEM with improved accuracy, spatial coverage and minimised voids. We have added this in the method section. This is being clarified in the supplementary information where detail account on DEM data has been addressed.

L318: Do these mesh sizes imply that you did not run the models using the original mesh size of a given DEM? If so, why? In addition, which resampling algorithm did you choose to change the grid resolution? The DEM in some of the figures looks a bit 'edgy', and I wonder if that the effect of a Nearest Neighbour interpolation.

Thanks for pointing out this concern. For the 30 DEMs, NASADEM and AW3D30 we used the default cell size, which is about 30 m, so no resampling was required. In r.avaflow this was achieved by leaving the cell size option blank. Also, in r.avaflow, cell sizes always have to be the cell size of the input DEM or a multiple of this cell size (e.g., for the 8 m DEM, it should be 8 m, 16 m, 24 m, 32 m, etc.). Thus, in order to mitigate this discrepancies in revised manuscript, we will conduct mesh size variations for only HMA-8m DEM and omit others for the parameter mesh size variation.

L322: Add 'the' or 'a' in front of GLOF.

Added 'the' as suggested.

L323: Remove 'into'

Removed as suggested.

L329: In my opinion, this is a really important point in the present manuscript: what is the volume of this lake? Table S2 suggests that the empirical equations alone differ by a factor of two (205 to 381 Mil m³), without accounting for uncertainties in the model parameters itself. What the equation yield is only the mean volume for a given lake area; the underlying data in lake-area-volume-relationships, however, may cover one to two orders of magnitude of estimated volumes for a given lake area. So, I wonder how representative these estimates are in order to provide a meaningful estimate of flood volume and discharge. It was also interesting to see that the volume from ice thickness models (last row in Table S2) almost triples the mean of means that you seem to use. In L338, you mention that you somehow adjusted the bathymetry, but how? Did you make the lake shallower? In any case, I would strongly encourage to also include a varying lake bathymetry in the variables that you assessed, as it remains unclear how much uncertainties in that variable propagate in your overall model result.

Thanks, reviewer, for your comments. We will add volume variation as one of the parameters for sensitivity analysis in the revised manuscript. We will accordingly amend the results and our conclusion.

L340-345: For non-experts, it might be good to show a histogram showing volumes of historic mass movements that generated GLOFs. This might help underline if these ten values cover a physically plausible range or not.

Thanks for the suggestions. We had the same concern for the variation of landslide volume, but we could not find any existing data sufficient to plot the histogram. We would readily incorporate it in our revised manuscript, if the reviewer has any suggested sources.

Figure S6: needs a color key that distinguishes the DEMs

Colour key has been added in the revised manuscript.

L354-355: So these six source locations alone give you 60 (6 locations x 10 volumes from 1 to 10 Mil m³) simulations? Or do you consider that all these sources produce mass movements simultaneously?

Thanks for providing this feedback. We want to clarify that we modelled one GLOF scenario from each location keeping the volume at $5 \times 10^6$ and all other input parameter values constant. To make this clear we amended lines 354 to 355 as "We then ran one GLOF scenario from each of these six locations, keeping all other input variables constant as defined in Table 2".

L371-379: Again some assumptions where we need more information: why can only the moraine provide material for entrainment? Isn't the broad floodplain downstream of Thorthormi Glacier full of sediments that can be entrained during a flood? In addition, what is the entrainment height? How did you measure the height of the moraine? How do you know that there isn't a bedrock sill, at which erosion might come to a halt during a GLOF?

We sincerely appreciate the reviewer's point. However, accurately measuring the depth and spatial extent of these erodible materials is highly challenging, even with field surveys, and remains infeasible using remote sensing techniques alone. In this study, we focused on the frontal moraine, as it is measurable using remote sensing data. The extent of the moraine was mapped using high-resolution Google Earth imagery, while its height was determined using the HMA-8m DEM. We have now added this information in section 3.2.5 (Lines 385 to 390).

L381-389: How much sense does it make to treat these variables independent of each other? I.e. only varying the internal friction angle, while keeping all other fixed? This is not really physically plausible, isn't it?

We appreciate the reviewer's concern and would like to clarify that r.avaflow does only offer the option to input each of these frictional parameters separately. The software offers no functionality to account for interdependencies between these parameters. Further, we are not aware of any robust empirical relationships between the tested parameters. Nevertheless, we

have provided clear recommendations for future studies to focus on the interactions of parameters, especially those we deemed highly sensitive as mentioned above.

L401: What are these 'outputs'?

Here we refer to output metrics including peak discharge, total discharge and flow arrival time, which we considered for the sensitivity analysis. We have made this clear in the revised manuscript.

L409: Isn't that standard deviation strongly dependent on the range of input values that you assessed? A narrower range in the input parameters will give you a smaller range in output values?

We agree that a narrower input parameter range would result in reduced variability and, consequently, a smaller standard deviation. However, the parameter value range used in this study is bookended by values widely reported in previously published literature. Therefore, we believe the range we employed is reasonable for each parameter. Additionally, we used the coefficient of variation (CV), calculated as the ratio of the standard deviation to mean. We believe this approach helps mitigate disparities arising from differences in the input parameter value range.

Figure 3: what do the dashed lines show? Typo in some panels: 'reolution'

The typo is corrected. The dashed line shows the flow of debris part of the GLOF. This is now mentioned in the caption of figure 3.

L445-446: Check grammar.

We have corrected the grammar to "The simulated flow reach distance was not sensitive to the mesh size: simulations with all three tested mesh sizes resulted in a flow reach distance of approx. 15 km".

L457: Please explain which volume you used in this simulation from the different sources (Loc-1 to Loc-6).

Thank you for pointing out this. We added the following statement as suggested: "We modelled a mass movement with a volume of $5 \times 10^6$ m$^3$ entering from the various locations we have identified in section 3.2.3."

L465: Peak discharge is measured m³ s-1. Do you mean 180 x 10³ m³ s-1?

Yes, 180 x 10³ m³ s$^{-1}$ is correct.

L466: 60,000 ³ s-1? I am really unsure about the values and the scale you show in Figure 4. How realistic is a peak discharge of 180000 ³ s-1? This would be one of the largest GLOF magnitudes in human history.

Thank you for pointing this out. We have now corrected all values and units. We believe that a peak discharge of 180 × 10³ m³ s$^{-1}$ is reasonable, given the area of Thorthormi Lake: its area is 5 km², is recently formed, and is the largest glacial lake in the HMA. Based on this area, we estimate its volume to be about 300 × 10⁶ m³. Assuming a mass movement of 5 × 10⁶ m³ enters the lake, this volume could trigger a GLOF event instantaneously due to the impact wave and subsequent moraine dam failure (contributing to the total flow) (Allen et al., 2022). Therefore, we believe that such a peak discharge value is feasible. As far as we are aware, no GLOF event of this magnitude has yet been reported from a lake of this size in the HMA. We cannot rule out a GLOF of such magnitude in the future due to changes in the cryosphere environment under warming conditions. We will address this concern in our planned future study which focuses on the downstream impact of the GLOF event from this lake.

Figure 5: Just to make sure that I understood it correctly: Linearly decreasing, and particularly negative, values mean that this parameter decreases the variance in the output? In other words, if you would still increase the value of this parameter, then your variance in GLOF discharge or volume would almost vanish?

No, a linearly decreasing slope shows the inverse relationship between the input parameter value variation and percentage change in model output parameters, not the variance. Yes, in the case of an inverse relationship, the in theory model result might approach zero if the input value is zero. However, this is not practically relevant, as the parameters are always defined within a reasonable range defined by the model writers and available in the previous literature as we adopted here.

Figure 6 and 7: sometimes you use double brackets )). Does that have a specific reason?

The double brackets are a typo. They are deleted in the revised manuscript.

L535: Word(s) missing at the end: GLOF … discharge, volume, arrival time?

Thanks. It is corrected.

L575: remove 'the' in front of 'multiple'

Amended as suggested.

L580: either 'datum' or 'dataset'

Amended as suggested.

L589: The effect of the DEM on GLOF output is indeed really interesting, but I could not follow your argument of river channel conveyance changing this output? What is this effect, could you explain this in more detail? I initially speculated that it's the lower surface roughness and friction stemming from coarser DEM resolutions that causes higher discharges? What is your take on this?

Thanks, we have amended line 589 in the revised manuscript:

"This limitation results in a reduction in surface roughness and river channel conveyance (carrying capacity of channel).  Thus, the flow spreads out more, leading to an increase in the modelled flow extent and reach".

L594-597: Interesting thought, do you have any evidence for this effect? Specifically why those changes might amplify GLOF magnitude?

We believe that it is reasonable to assume that topographic features may change over time, as evidenced by Khosh Bin Ghomash et al. (2019), Bishop et al. (2002) and Watson et al. (2015). While we do not necessarily suggest that such changes will amplify the GLOF, they will have substantial influence on its flow characteristics. This introduces huge uncertainty if a DEM with inadequate spatial and temporal resolution is used in GLOF modelling. To make this clear, we have added discussion in line 618 to 620.

L604: rephrase to 'a co-registration algorithm developed by Shean et al. (2016) or so?

Thanks. Amended as suggested.

L611: 'DEMs'

Amended as suggested.

L615-617: A problem with this conclusion is that you inherently infer that the simulations using the HMA-DEM are better/ more realistic or whatsoever, as they produce smaller discharges. However, as you do not provide any validation/ reference dataset, it is difficult to judge if one DEM really outperforms the others. You also have no independent validation dataset to show that the HMA-DEM has fewer errors than the others, nor do you show how the noise in these DEMs propagates in your simulations.

As correctly pointed out by the reviewer, in this study neither we want to conclude that HMA-8m DEM is a better option nor want to recommend the specific name of the DEM for future studies as our study does not support this conclusion. Instead, our goal is to emphasize using DEM of good quality for GLOF modelling in terms of both spatial and temporal coverage. To avoid confusion, we have added "HMA-8m DEM" in the discussion from line 615 to 620.

L646: To be fair, you also did not explicitly mention the angles and directions of these mass movements anywhere in your manuscript.

Thanks for pointing out this. As suggested, we have mentioned here the angles and directions of mass movements entering the lake from various locations as all locations we identified have different angles and direction concerning the lake. However, this information was not mentioned clearly in the method and result section. We will mention directions and angles clearly in our revised manuscript.

L659: A provocative conclusion would be that you can obtain highly contrasting GLOF discharges for the same lake, just by moving the initiation zone of your mass movement to another location of the slope. How much sense do these worst-case scenarios make then, if there is – in theory - infinite combinations of source location, volume, velocity, etc. of a landslide entering a lake?

Thank you. While we agree that, in theory, there are infinite combinations of parameters contributing to uncertainties, there is no silver bullet solution to pinpoint the exact source or quantify the precise volume of mass movements entering the lake in the foreseeable future to eliminate this uncertainty. However, we believe that by identifying and quantifying the uncertainties related to the origin of mass movements in addition to the volume, we believe it is reasonable to propose considering these origins of mass movements in designing the scenario-based GLOF modelling. This approach goes beyond the current practice, which primarily focuses on the volume of mass movements entering the lake. This is important because, as demonstrated in our study, the impact of mass movements can vary significantly depending on its origin. In some cases, what may be considered a worst-case scenario from

one origin might represent only a low-magnitude event from another. We have discussed this issue in discussion (line 662 to 669)  for more clarity.

L661-662: I am not sure whether this conclusion is valid. You modelled the effect of increasing landslide size only for the case with the highest consequences (loc-1). Would you expect similar effects on GLOF peak discharge, if you were to model landslide impacts from the other locations, say loc-3, where the wave might be dampened as it is first pushed against the opposite valley wall?

We agree that there is a bias when comparing mass movement volume with grain density, but we believe it is logical to put them together as we are trying to convey that volume plays greater role than the grain density, as both of them are the characteristics of mass movement entering lake. Yes, there will be obvious difference in resulting GLOF if we model mass movement from the different locations but that too will also depend on volume and grain density. Having said this, we believe that impact on GLOF magnitude due to uncertainties in origin of avalanche is effectively addressed by our one of the parameters that consider avalanche from various locations.

L697: How would one measure δ in the field? Could you advise? And what is a statistically substantial sensitivity analysis and how would one do that?

Thanks for pointing out this. It is practically not possible to measure a basal friction angle at the base of a landslide moving at a velocity of 100 or 200 km/h. In practice, suitable values are usually derived through back-calculations (Mergili et al., 2018). Thus , we have corrected our statement (in lines 724 to 727) as:

"While the back calculated values might seems reasonable initiation value for basal friction angle as measuring it in the field practically not feasible we recommend conducting a statistically substantial sensitivity analysis using adequate sampling size and an appropriate statistical model."

L700: Verb missing.

The missing verb is added.

L717: How should that 'careful treatment' be done? By using a range of values and aggregating the results? Many papers claim that the research was done with utmost care – here you have space to explain and give recommendations how to approach these uncertainties in GLOF modelling.

We believe that this is now effectively addressed by our response to the previous comments about recommendations for how each parameter should be treated

L722: Check grammar.

Corrected. Thanks

L729: You often mention that results should be interpreted in caution. I agree, but what exactly is it that make you caution against these results and what is your suggestion to move forward? Users might want to have some guideline how to attain a certain level of confidence in their r.avaflow model results.

Thanks for pointing out this. We have now changed this statement to "but the modelling results can be subjected to substantial uncertainty". We will do same for all appearances and provide clear guidelines wherever necessary.

758: one 'arrival and' used too often?

Corrected the typo. Will do the same for any other typos.

L770-781: That paragraph offers no new insights and can be deleted entirely in my opinion.

Thanks for the suggestions. We have now deleted this paragraph.

L788-789: Again, I feel a bit stranded with this note of caution. In which settings do you expect the tested parameters to be substantially different from your setting?

Thanks to the reviewer for this feedback. Here intent is to clarify that the model parameters tested here do not necessarily represent the parameters in other models excluding r.avaflow we used here. To make this clear we have amended it as follows: "However, it should be noted we all the parameters tested here do not necessarily apply to all models used for GLOF simulations".

L800-801: It is really surprising to me that this study did not assess any parameter interaction, especially as you point at the strong effects of some of the parameters. What would happen with flood volume and discharge, if a user selects a very large DEM resolution, a high landslide volume, and a high entrainment coefficient? Will you just get super high GLOF outputs or do you expect them to cancel out each other? I understand that your sensitivity analysis seeks to model the output by varying one parameter and keeping all others constant; however, I would

be delighted to see some kind of recommendation, which parameter values might be suitable for a first try, and in particular, what would be a really bad initial set of parameters in GLOF forward modelling.

We appreciate the genuine concern shared by the reviewer about the effect the interaction of parameters will have on the modelling results. Please refer to our earlier response about the challenges of modelling for all parameter interactions.

References

ALLEN, S. K., SATTAR, A., KING, O., ZHANG, G., BHATTACHARYA, A., YAO, T. & BOLCH, T. 2022. Glacial lake outburst flood hazard under current and future conditions: worst-case scenarios in a transboundary Himalayan basin. *Natural Hazards and Earth System Sciences,* 22**,** 3765-3785.

BAGGIO, T., MERGILI, M. & D'AGOSTINO, V. 2021. Advances in the simulation of debris flow erosion: The case study of the Rio Gere (Italy) event of the 4th August 2017. *Geomorphology,* 381.

BISHOP, M. P., SHRODER, J. F., BONK, R. & OLSENHOLLER, J. 2002. Geomorphic change in high mountains: a western Himalayan perspective. *Global and Planetary Change,* 32**,** 311-329.

BYERS, A. C., ROUNCE, D. R., SHUGAR, D. H., LALA, J. M., BYERS, E. A. & REGMI, D. 2018. A rockfall-induced glacial lake outburst flood, Upper Barun Valley, Nepal. *Landslides,* 16**,** 533-549.

KHOSH BIN GHOMASH, S., CAVIEDES-VOULLIEME, D. & HINZ, C. 2019. Effects of erosion-induced changes to topography on runoff dynamics. *Journal of Hydrology,* 573**,** 811-828.

LALA, J. M., ROUNCE, D. R. & MCKINNEY, D. C. 2018. Modeling the glacial lake outburst flood process chain in the Nepal Himalaya: reassessing Imja Tsho's hazard. *Hydrology and Earth System Sciences,* 22**,** 3721-3737.

MERGILI, M., FRANK, B., FISCHER, J.-T., HUGGEL, C. & PUDASAINI, S. P. 2018. Computational experiments on the 1962 and 1970 landslide events at Huascarán (Peru) with r.avaflow: Lessons learned for predictive mass flow simulations. *Geomorphology,* 322**,** 15-28.

MERGILI, M., PUDASAINI, S. P., EMMER, A., FISCHER, J.-T., COCHACHIN, A. & FREY, H. 2020. Reconstruction of the 1941 GLOF process chain at Lake Palcacocha (Cordillera Blanca, Peru). *Hydrology and Earth System Sciences,* 24**,** 93-114.

NCHM 2019. Detailed assessment report on GLOF hazard from Thorthormi glacial lakes and asssociated glaciers. Thimphu.

ROUNCE, D. R., MCKINNEY, D. C., LALA, J. M., BYERS, A. C. & WATSON, C. S. 2016. A new remote hazard and risk assessment framework for glacial lakes in the Nepal Himalaya. *Hydrology and Earth System Sciences,* 20**,** 3455-3475.

VILCA, O., MERGILI, M., EMMER, A., FREY, H. & HUGGEL, C. 2021. The 2020 glacial lake outburst flood process chain at Lake Salkantaycocha (Cordillera Vilcabamba, Peru). *Landslides,* 18**,** 2211-2223.

WATSON, C. S., CARRIVICK, J. & QUINCEY, D. 2015. An improved method to represent DEM uncertainty in glacial lake outburst flood propagation using stochastic simulations. *Journal of Hydrology,* 529**,** 1373-1389.

ZHANG, T., WANG, W. & AN, B. 2024. A massive lateral moraine collapse triggered the 2023 South Lhonak Lake outburst flood, Sikkim Himalayas. *Landslides,* 22**,** 299-311.

ZHENG, G., MERGILI, M., EMMER, A., ALLEN, S., BAO, A., GUO, H. & STOFFEL, M. 2021. The 2020 glacial lake outburst flood at Jinwuco, Tibet: causes, impacts, and implications for hazard and risk assessment. *The Cryosphere Discuss.,* 2021**,** 1-28.